# Serum proteomics links suppression of tumor immunity to ancestry and lethal prostate cancer

Tsion Zewdu Minas[1,12], Julián Candia [1,2,12], Tiffany H. Dorsey[1], Francine Baker [1], Wei Tang [1],
Maeve Kiely [1], Cheryl J. Smith[1], Amy L. Zhang[1], Symone V. Jordan[1], Obadi M. Obadi[1], Anuoluwapo Ajao[1],
Yao Tettey[3], Richard B. Biritwum[3], Andrew A. Adjei[3], James E. Mensah[3], Robert N. Hoover[4], Frank J. Jenkins[5],
Rick Kittles [6], Ann W. Hsing[7,8], Xin W. Wang[1,9], Christopher A. Loffredo[10], Clayton Yates [11],
Michael B. Cook[4] & Stefan Ambs [1✉]

There is evidence that tumor immunobiology and immunotherapy response may differ between African American and European American prostate cancer patients. Here, we determine if men of African descent harbor a unique systemic immune-oncological signature and measure 82 circulating proteins in almost 3000 Ghanaian, African American, and European American men. Protein signatures for suppression of tumor immunity and chemotaxis are elevated in men of West African ancestry. Importantly, the suppression of tumor immunity protein signature associates with metastatic and lethal prostate cancer, pointing to clinical importance. Moreover, two markers, pleiotrophin and TNFRSF9, predict poor disease survival specifically among African American men. These findings indicate that immune-oncology marker profiles differ between men of African and European descent. These differences may contribute to the disproportionate burden of lethal prostate cancer in men of African ancestry. The elevated peripheral suppression of tumor immunity may have important implication for guidance of cancer therapy which could particularly benefit African American patients.

[1] Laboratory of Human Carcinogenesis, Center for Cancer Research, National Cancer Institute, National Institutes of Health, Bethesda, MD, USA.
[2] Longitudinal Studies Section, Translational Gerontology Branch, National Institute on Aging, National Institutes of Health, Baltimore, MD, USA. [3] University of Ghana Medical School, Accra, Ghana. [4] Division of Cancer Epidemiology & Genetics, National Cancer Institute, National Institutes of Health, Bethesda, MD, USA. [5] Hillman Cancer Center, University of Pittsburgh, Pittsburgh, PA, USA. [6] Division of Health Equities, Department of Population Sciences, City of Hope Comprehensive Cancer Center, Duarte, CA, USA. [7] Stanford Cancer Institute, Stanford School of Medicine, Palo Alto, CA, USA. [8] Stanford Prevention Research Center, Stanford School of Medicine, Palo Alto, CA, USA. [9] Liver Cancer Program, Center for Cancer Research, National Cancer Institute, National Institutes of Health, Bethesda, MD, USA. [10] Cancer Prevention and Control Program, Lombardi Comprehensive Cancer Center, Georgetown University Medical Center, Washington, DC, USA. [11] Department of Biology and Center for Cancer Research, Tuskegee University, Tuskegee, AL, USA. [12] These authors contributed equally: Tsion Zewdu Minas, Julián Candia. ✉email: ambss@mail.nih.gov

Men of African origin bear the highest prostate cancer burden in the U.S. and globally[1–3]. They are at an increased risk of developing fatal prostate cancer in the U.S and England[4] and present with more aggressive disease in the Caribbean and sub-Saharan Africa[2,5]. The reasons for the observed global prostate cancer health disparities are unclear but are likely related to an array of factors such as access to health care, lifestyle and environment, and ancestral and biological factors[6–8].

Previously, we and others described that tumor immunobiology differs between African-American (AA) and European-American (EA) prostate cancer patients[9–12]. A tumor-specific immune-inflammation gene expression signature was more prevalent in prostate tumors of AA than EA patients[11]. The occurrence of this signature in prostate tumors was associated with decreased recurrence-free survival[13]. Furthermore, regular use of aspirin, an anti-inflammatory drug, may reduce the risk of aggressive prostate cancer, disease recurrence and lethal disease in AA men[14,15]. Combined, these findings suggest that inflammation and host immunity may contribute to prostate cancer progression but with notable differences between AA and EA men.

Ancestral factors can influence immune-related pathways[16]. Germline genetic variant prevalence and alternative splicing in immune-inflammation-related genes can show large differences amongst population groups[17–19]. Hence, the immune-inflammation gene expression signature identified in the tumors of AA prostate cancer patients could be due to either tumor biology and the associated microenvironment, ancestral factors, or systemic differences in immune-oncology marker expression.

In this work, we test the hypothesis that a distinct systemic immune-inflammation signature exists in men of African ancestry that associates with prostate cancer using a large cohort of diverse men. Applying large-scale proteomics with Olink technology, we discover the up-regulation of circulating immune-oncological proteins that functionally relate to chemotaxis and suppression of tumor immunity and their association with West African ancestry and lethal prostate cancer. Our findings point to the clinical importance of a serum proteomic signature in prostate cancer patients that may affect men of African ancestry more so than other men.

## Results

### Large-scale evaluation of immune-oncological proteins in the NCI-Maryland and NCI-Ghana prostate cancer studies.

To investigate if men of African descent are differentially affected by a systemic immune inflammation, we utilized two case-control studies with large representations of men of African ancestry: the NCI-Ghana and NCI-Maryland Prostate Cancer Case-Control Studies. Characteristics of the participants in the two studies have been previously described[14,20]. For our investigation, limited to men with quality-controlled (QCed) serum proteome measurements, the NCI-Ghana Prostate Cancer Case-Control study included 1143 men of whom 489 were cases and 654 were controls. Cases were older than controls (Supplementary Table 1). Controls had a slightly lower body-mass index (BMI) than cases. In addition, more controls were current smokers than cases (14% vs. 2%). High Gleason score (>7) was reported in 158 out of the 489 patients (32%). Participants in the NCI-Maryland Prostate Cancer Case-Control study included 1647 AA and EA men of whom 819 were cases and 828 were controls. Cases were slightly younger than controls. Cases and controls had similar BMI distributions. In addition, more cases were current smokers than controls (24% vs. 14%). High Gleason score (>7) was reported in 141 out of the 819 patients (17%). In the control population, Ghanaian men tended to be younger, to have a lower BMI, less

likely to be diagnosed with diabetes, but with a similar prevalence of current smokers, when compared with AA and EA men.

We assayed 92 circulating immune-oncological proteins in a total of 3094 serum samples containing 1505 controls and 1432 cases along with 157 randomly selected blinded duplicates. To control for any batch effects, the serum samples were assayed in a random order along with the 5% blind duplicates for intensity normalization (see Methods). Ninety-five percent of the samples passed stringent quality control, leaving 1482 controls (654 Ghanaian, 374 AA, and 454 EA) and 1308 cases (489 Ghanaian, 394 AA, and 425 EA) for our analysis (Supplementary Table 1). The average intra- and inter-plate coefficients of variation calculated based on duplicates were very low at 1.7% and 2.6%, respectively. In addition, the proportion of variance explained by an inter-plate batch effect was rather minimal for each of the serum proteins even before intensity normalization (Supplementary Fig. 1). Out of the 92 serum proteins, 61 had abundance levels above the lower limit of detection (LLOD) in 100% of the samples tested (Supplementary Table 2, Supplementary Fig. 2) and 78 proteins had levels above LLOD in >50% of the samples (Supplementary Fig. 2). Because 10 out of the 92 serum proteins were detected (i.e., had levels above LLOD) in <20% of the samples (Supplementary Fig. 2), only the remaining 82 proteins were included in our analysis (Supplementary Table 3). Next, we assessed how the 82 serum markers correlate with one another in Ghanaian, AA, and EA men with and without prostate cancer using Pearson's pairwise correlation analysis (Fig. 1 and Supplementary Fig. 3). The top ten observed correlations for each population group are presented in Supplementary Table 4. Most of these relationships have not previously been described. Notably, epidermal growth factor levels strongly correlated with CD40L, a marker of activated T cells, in cases [AA ($r = 0.87$), and EA ($r = 0.77$)] and controls [Ghanaian ($r = 0.71$), AA ($r = 0.83$), and EA ($r = 0.80$)]. Interleukin-8 levels correlated consistently with circulating caspase-8 in cases [Ghanaian ($r = 0.48$), AA ($r = 0.74$), and EA ($r = 0.73$)] and controls [Ghanaian ($r = 0.69$), AA ($r = 0.82$), and EA ($r = 0.80$)]. Other robust correlations include interleukin-8 with MCP3, TNFRSF4 with TNFRSF9, and CD83 with TNFRSF9.

### Clinical and socio-demographic characteristics are associated with immune-oncological proteins.

Cytokine levels can be influenced by environmental exposures and disease. Therefore, we investigated the association between various socio-demographic and clinical characteristics (age, BMI, education, aspirin use, smoking, diabetes, and PSA) with serum levels of immuno-oncological proteins using a multivariable linear regression model with adjustment for multicomparison analysis (Fig. 2). We restricted this analysis to the control population in the NCI-Ghana and NCI-Maryland studies to exclude the potential confounding effect of prostate cancer in the analysis. Among the exposures, aspirin use, and blood PSA levels showed only few relationships with the profile of the 82 immune-oncology markers. Other exposures and several demographics showed more robust relationships.

Aging is known to impact the immune system and is a risk factor for many diseases including cancer[21]. In our analysis, aging was most consistently associated with the level of the analytes across the three population groups, showing a significant correlation with almost half of these circulating immune-oncological proteins. For example, PGF, CXCL9, Gal9, Gal1, CX3CL1, TNFRSF12A, CCL23, MMP7, DCN, MMP12, ADGRG1, and PTN positively associated with age in all three population groups. The top-ranked biological functions that associated with these age-related proteins were cell migration and

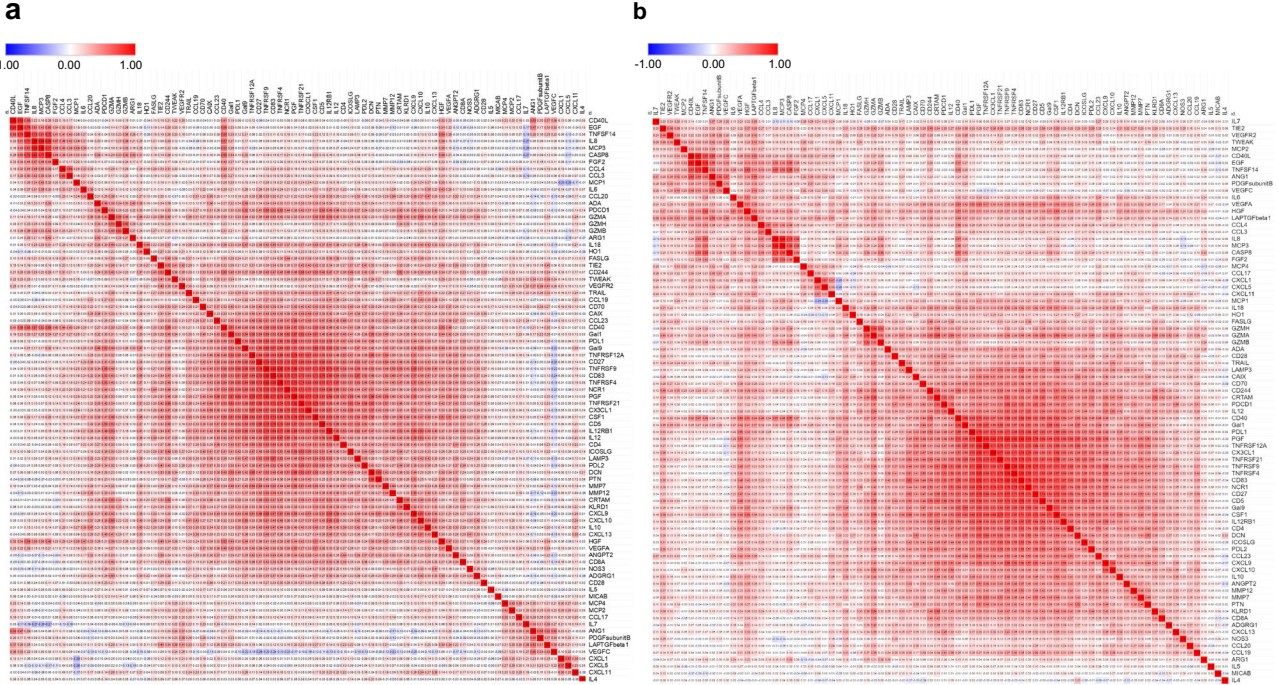

**Fig. 1 Correlation matrix presenting Pearson pairwise correlations for each of the 82 serum protein pairs in African American men.** Pearson pairwise correlations were estimated for each serum protein pair in African American (**a**) population controls (*n* = 374) and (**b**) prostate cancer cases (*n* = 394). Source data are provided as a Source Data file.

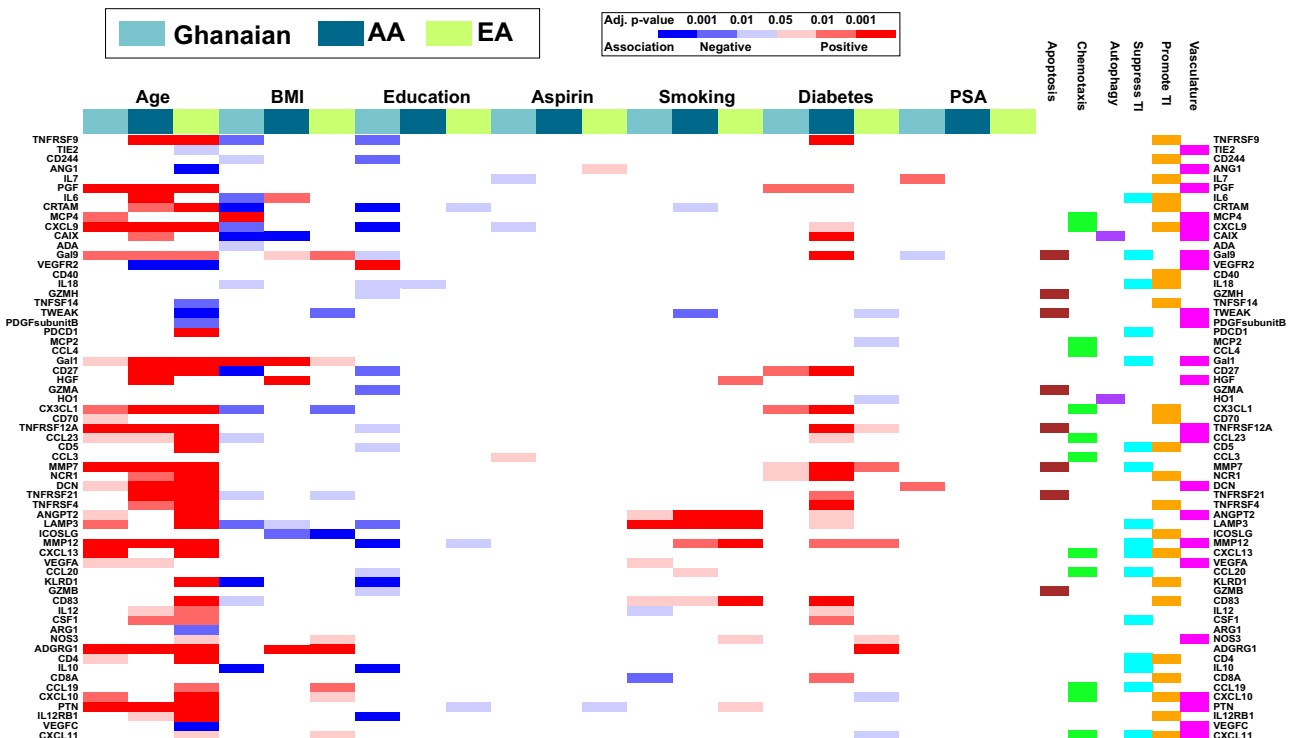

**Fig. 2 Association of socio-demographic and clinical characteristics with systemic immune-oncological proteins in Ghanaian (*n* = 654), AA (*n* = 374), and EA (*n* = 454) men without prostate cancer.** The association of the 82 immuno-oncological proteins (as continuous variables) with age, BMI, education, aspirin use, smoking, diabetes, and PSA was assessed in men without prostate cancer using a multivariable linear regression test. *P* values were adjusted for multiple comparison. An analyte was considered significantly associated with clinical and socio-demographic covariables if the multivariable model yielded a false discovery rate (FDR)-adjusted *P* < 0.05 on the F-statistic. Analytes that did not have a significant association with any of the clinical/ sociodemographic variables in at least one of the population groups are not presented in the heatmap. Blue represents a negative association while red represents a positive association. The significance level (FDR-adjusted two-sided *P* value-based) for each association is color-coded. Source data are provided as a Source Data file. TI tumor immunity, AA African American, and EA European American.

positive regulation of cell communication (Supplementary Fig. 4). Lastly, TNFRSF9, CD27, TNFRSF21, TNFRSF4, and IL12RB1 were positively associated with age while VEGFR2, a tyrosine kinase receptor for VEGF, was negatively associated with age in EA and AA men (Fig. 2 and Supplementary Data 1).

In contrast to the positive association of many of the immune-oncological proteins with age, BMI tended to be negatively associated with these circulating immune-oncological analytes. For instance, 16 of the immune-oncological proteins negatively associated with BMI among Ghanaian men (Fig. 2). This finding may be surprising as obesity is generally thought to be associated with systemic inflammation. On the contrary, serum GAL1, a glycan binding protein that mediates the suppressive function of $T_{Reg}$ cells[22], showed the opposite trend and was positively associated with BMI in all three population groups.

To explore how the social/behavioral environment may affect immune-oncological serum protein levels, we investigated their relationship with educational attainment. For Ghanaian men, 18 of the 82 immuno-oncological markers were negatively associated with their education level (Fig. 2). Among EA men, three of the 82 immune-oncological proteins had significant inverse relationships with the attained level of education (Fig. 2), with two of these markers showing a similar pattern among Ghanaian and EA men.

Previous studies have shown that tobacco smoking increases inflammation[23]. Herein, we assessed the association between cigarette use (never, former, vs. current smoker) on the level of immune-oncological proteins in circulation. We found that current smoking was consistently associated with significantly increased level of analytes that regulate angiogenesis (ANGPT2), antigen presentation (CD83), and autophagy (LAMP3), in all three study populations (Fig. 2).

Innate immune system-driven inflammatory processes have been implicated in the pathogenesis of diabetes[24]. In our analysis, among the proteins that showed an association with self-reported diabetes, a matrix metalloprotease enzyme, MMP7, was positively associated with diabetes in all three population groups (Fig. 2). On the other hand, CX3CL1 was positively associated with diabetes exclusively in men with African ancestry (Fig. 2 and Supplementary Data 1). CX3CL1 is known to regulate insulin secretion[25], is elevated in the serum of patients with type 2 diabetes[26], and has been implicated in diabetic nephropathy[27], validating the findings in our study.

C-reactive protein (CRP) is a commonly measured pro-inflammatory marker in the body and has been reported to be associated with worse prostate cancer prognosis[28,29]. Because it was not part of our marker panel, we measured blood CRP in 156 plasma samples from population controls of the NCI-Maryland study. Smoking was the only socio-demographic variable that showed association with CRP, however, the observed association was lost when adjusted for multiple testing (Supplementary Table 5). Furthermore, CRP showed positive associations with 24 of the 82 serum proteins (TNFRSF9, IL7, PGF, IL6, Gal9, GZMH, CXCL1, TNFSF14, Gal1, PDL1, HGF, HO1, CD70, TNFRSF12A, CCL3, MMP7, ANGPT2, VEGFA, CCL20, KLRD1, CSF1, CD4, MCP3, and CXCL11).

**The systemic immune-oncological cytokine profile in men of African ancestry is distinct from men of European ancestry.** To investigate if ancestral population group differences may influence circulating levels of the immune-oncological markers, we performed an unsupervised clustering analysis examining how the levels of the 82 immune-oncological analytes would group men without prostate cancer from Ghana and the US. Notably, these analytes tended to cluster by population group, with levels in Ghanaian men being most distant from EA men while AA samples tended to cluster in between these two groups (Fig. 3), suggesting that the ancestral background may have a significant impact on the global immune-oncological protein profile. We performed an additional statistical analysis of cluster assignments to more formally establish that the immune-oncological protein profile defined by the 82 markers is indeed different between these groups of men. We obtained the cluster assignments by cutting the hierarchical clustering dendrogram to extract K clusters (with K = 2, 3) and tested for differences in their distribution across the population groups (Supplementary Fig. 5). We found significant differences in cluster representation between Ghanaian, AA, and EA men with cluster enrichment by population group at $P < 1.e-10$, confirming that significant differences likely exist in the global immune-oncological protein profile among them.

To further evaluate the influence of ancestry, we estimated West African ancestry in AA and EA population controls of the NCI-Maryland study and its relationship with the cytokine profile. West African ancestry was determined using 100 validated ancestry informative markers[30]. The approach showed that, to some extent, the variance in the levels of several immune-oncological analytes can be strongly influenced by the degree of West African ancestry of these individuals (Fig. 4a). The variance in 39 of the analytes were significantly [false discovery rate (FDR)-adjusted $P < 0.05$] influenced by degree of West African ancestry (Supplementary Table 6, Supplementary Data 2). The levels of 37 analytes were significantly accounted for by West African ancestry even after adjusting for age, BMI, aspirin use, education, income, diabetes, and smoking status (Supplementary Table 7, Supplementary Data 3). CXCL5, CXCL1, MCP2, MCP1, CXCL11, CCL23, PTN, TWEAK, NCR1, IL18, and CCL17 were the top-ranked proteins. West African ancestry contributed to the variance with various effect sizes and explained >10% of the variance among the top 7 proteins (Supplementary Tables 6–7, Supplementary Data 2–3). For instance, 41% and 50% of the variance in the serum levels of CXCL1 and CXCL5, respectively, was accounted for by the degree of West African ancestry (Fig. 4a, Supplementary Tables 6–7, Supplementary Data 2–3). When we compared the levels of these proteins across the three population groups, we observed a significant African ancestry-related trend (Fig. 4b), with 10 of the 82 circulating immune-oncological proteins (CXCL5, CXCL1, CXCL11, MCP2, CCL17, MCP4, CD70, PDL2, MMP7, and CCL19) being significantly elevated in both Ghanaian and AA men compared to EA men (Supplementary Table 8); 13 other markers (MCP1, IL12, CCL23, CD8A, NCR1, TNFRSF4,TNFSF14, TWEAK, IL7, HGF, HO1, TNFRSF21, and ANG1) were inversely related to West African ancestry (Supplementary Table 9).

**Cytokines associated with suppression of tumor immunity and chemotaxis are upregulated in men of African ancestry.** Levels of many of the 82 immune-oncology markers showed a marked association with ancestry. To better define the functional implications of these population group differences, we grouped the 82 proteins into six biological processes according to Olink guidelines (Supplementary Table 10): apoptosis/cell killing, autophagy/metabolism, chemotaxis/trafficking to tumor, suppression of tumor immunity (Th2 response, tolerogenic), promotion of tumor immunity (Th1 responses), or vasculature and tissue remodeling. To gain insight on how activation of these six processes/pathways may differ by population group, we compared process/pathway sum scores between Ghanaian, AA, and EA men without prostate cancer using multicomparison-adjusted significance testing. Of these pathways, chemotaxis, promotion of

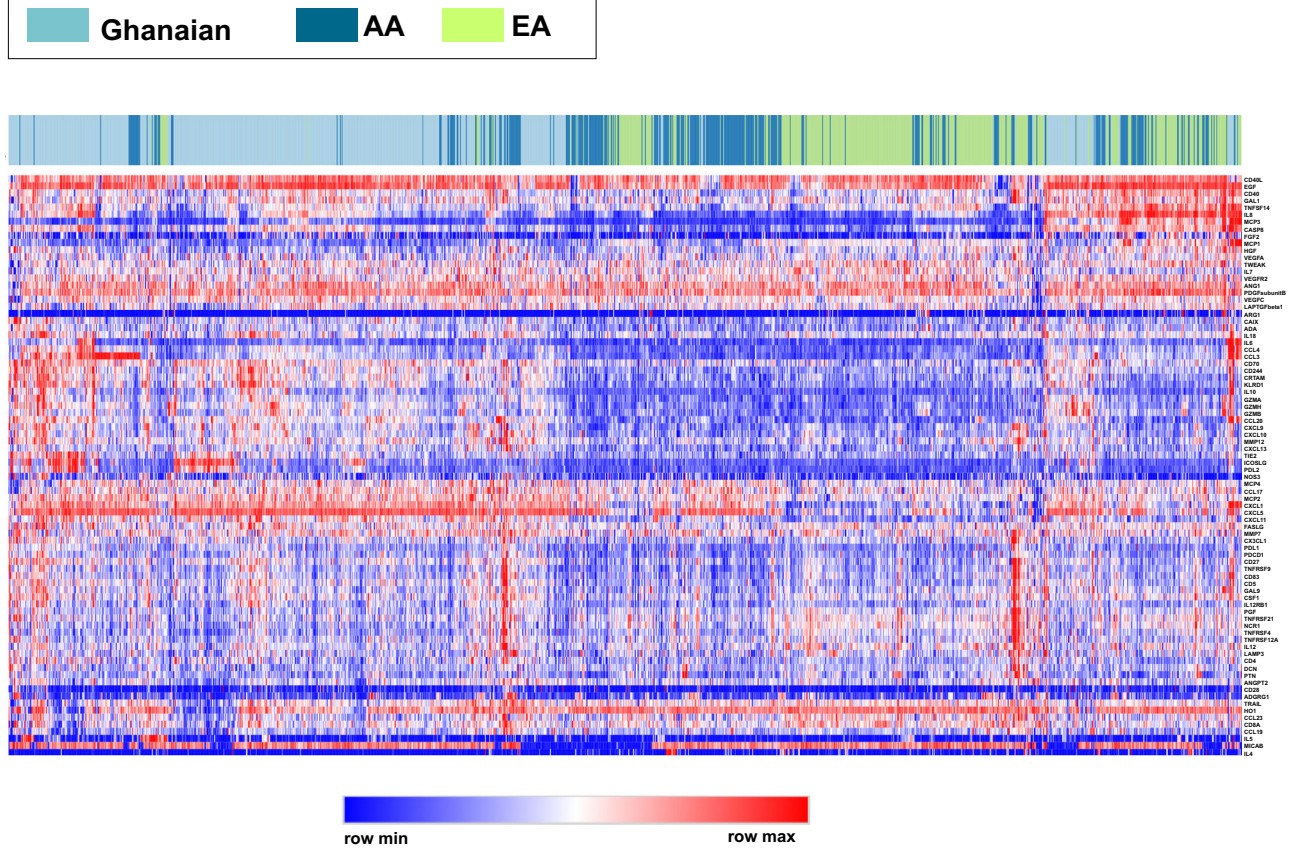

**Fig. 3 Unsupervised hierarchical clustering associates circulating immune-oncological proteome profiles with population groups—Ghanaian, AA, and EA men.** Heatmap showing protein profiles for men without prostate cancer. Each row represents a protein ($n = 82$), and each column corresponds to an individual [$n = 1482$ (654 Ghanaian, 374 AA, and 454 EA)]. Each individual is color-coded as Ghanaian, AA, or EA in the annotation bar on top of the heatmap. Normalized $z$-score of proteins abundance is depicted on a low-to-high scale (blue-white-red). Source data are provided as a Source Data file. AA African American and EA European American.

tumor immunity, and suppression of tumor immunity were different in their predicted activity between AA and EA men (Fig. 5). AA men had higher scores for chemotaxis and suppression of tumor immunity when compared to EA men, indicating higher pathway activity in AA men, but they had a lower score for promotion of tumor immunity. Ghanaian men had even higher scores for chemotaxis and suppression of tumor immunity than both AA and EA men (Fig. 5c, e), indicating a possible association with West African ancestry. The latter was corroborated with our finding that the chemotaxis and suppression of tumor immunity scores positively correlated with the proportion of West African ancestry within the NCI-Maryland cohort, even after holding the other variables constant (i.e., age, BMI, education, aspirin use, diabetes, and smoking history) in the regression analysis (for chemotaxis score: regression coefficient = 5.12 (3.75, 6.49), $P < 0.0001$; for suppression of immunity score: regression coefficient = 4.02 (2.01, 6.04), $P < 0.0001$). Even though apoptosis and vasculature-associated cytokines were not significantly different between EA and AA men, we found both processes to be elevated in the Ghanaian men.

**Suppression of tumor immunity is associated with reduced survival of prostate cancer patients.** Next, we examined the clinical implication of our findings and assessed the association of pathway activity with survival of prostate cancer patients or controls in the NCI-Maryland study using multivariable Cox regression modeling. As of the end of 2018, out of the 819 cases, there have been 202 deaths in our case population, of whom 103

(51%) had a cancer diagnosis as the recorded primary cause of death, and 28% of all deaths ($n = 57$) were directly attributed to prostate cancer. On the other hand, 99 of the 828 population controls had died by the end of 2018. Median survival follow-up for cases and controls were 8.6 and 6.7 years, respectively. With these data, we built a Cox regression model with the six biological processes/pathways and additional adjustments for other covariables including disease status (see Methods). Among the six pathways, only suppression of tumor immunity showed independent association with survival outcomes among the cases in a disease status-adjusted analysis (i.e., PSA, TNM Stage, Gleason score, and Gleason pattern) defined by the National Comprehensive Cancer Network (NCCN) risk score (Fig. 6 with 99% CI, Supplementary Fig. 6 with 95% CI). Prostate cancer patients with an increased activity of this pathway had the highest risk of death from all causes (Fig. 6a, Supplementary Table 11). In contrast, suppression of tumor immunity was not associated with all-cause mortality of population controls (Supplementary Table 12), suggesting that the association with all-cause mortality among cases might be prostate cancer-related. Prostate cancer patients with elevated suppression of tumor immunity at diagnosis had also the highest risk of a prostate cancer-specific mortality, although statistically significant only with a 95% CI (Supplementary Fig. 6), but not with a multicomparison-adjusted 99% CI (Fig. 6b, Supplementary Table 13). Lastly, prostate cancer patients with increased suppression of tumor immunity were also significantly more likely to die from any cancer (prostate cancer or secondary cancer) following the prostate cancer diagnosis (Fig. 6c,

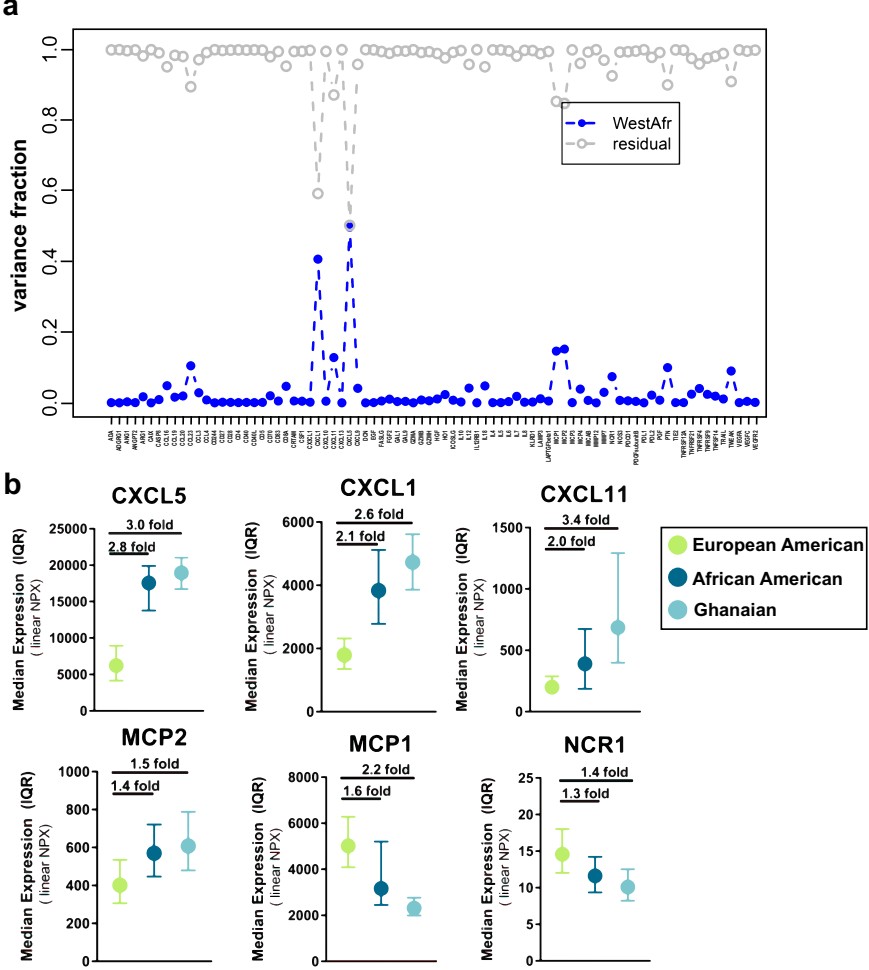

**Fig. 4 Immune-oncological proteins and their relationship with West-African ancestry. a** Variance analysis for the levels of each of the 82 immune-oncological cytokines assessed as a function of genetic estimation of West African admixture among men without prostate cancer within the NCI-Maryland study ($n = 795$). The blue plot represents the proportion of variance that can be explained by the degree of West-African admixture while the grey plot represents the residual variance that remains to be explained by factors other than West-African ancestry. **b** The median levels of the top six West-African ancestry-correlated immune-oncological proteins were compared between Ghanaian ($n = 654$), AA ($n = 374$), and EA ($n = 454$) men. Error bars represent inter quartile range (IQR). Linearized protein abundances (2^NPX) were used to determine median and IQR for each of the proteins. Data are presented as median ± IQR. Source data are provided as a Source Data file.

Supplementary Table 14), indicating a more general predisposition to cancer in patients with a high suppression of tumor immunity score in this hypothesis generating analysis approach.

**Elevated suppression of tumor immunity is associated with metastatic prostate cancer.** To further corroborate the significance of suppression of tumor immunity in the development of lethal prostate cancer, we assessed its association with prostate cancer aggressiveness per NCCN guidelines (see Methods). Information on TNM stage was only obtainable for the NCI-Maryland prostate cancer patients, hence only these cases were scored according to the NCCN guidelines. Patients with a high suppression of tumor immunity score were at substantially increased odds of being diagnosed with regional or distant metastasis (OR 3.79, 95% CI 1.59–9.04, >median vs. ≤median) (Table 1), consistent with the disease survival data. The data showed a significant trend in the association of elevated suppression of tumor immunity with disease aggressiveness ($P$ trend = 0.004) (Table 1). Although a stratified analysis by self-reported race/ethnicity suggested that high suppression of tumor immunity was associated with metastatic prostate cancer more strongly among AA than EA men, large 95% CIs for the odds

ratio precluded the observed difference across racial groups from statistical significance.

**Blood levels of TNFRSF9/CD137/4-1BB and pleiotrophin predict lethal prostate cancer among AA men.** To identify potential drivers of the relationship between immune-oncology markers and lethal prostate cancer, we applied a cross-validated, regularized Cox regression model using eNetXplorer (see Methods). Included in this model were the 82 immune-oncology markers and 6 patient feature covariates (age, education, BMI, smoking history, aspirin use, and diabetes). Of those patient features, education as a surrogate for socioeconomic status and health care access, BMI, smoking status, and aspirin use have previously been associated with the risk of lethal prostate cancer whereas the direction and strength association for diabetes with prostate cancer and disease outcomes is more uncertain[31,32]. Utilizing this method, we could not identify a robust predictive signature of lethal prostate cancer for EA patients. However, for AA patients, lasso regression (alpha = 1) was selected as a predictive model ($P = 0.0001$) with the best overall performance across alpha values (Supplementary Fig. 7A, Supplementary Data 4). In this model, a signature consisting of TNFRSF9 and

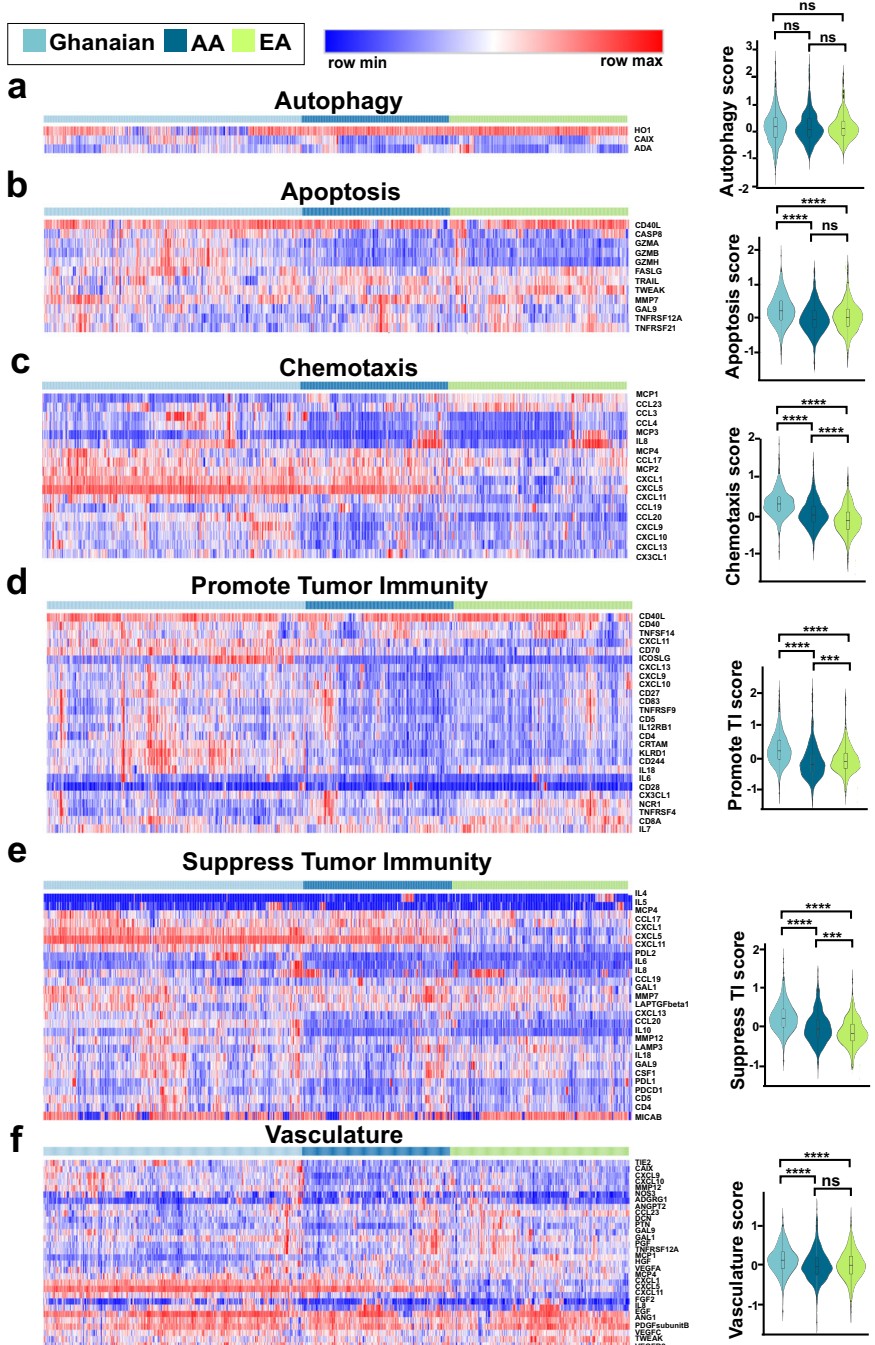

**Fig. 5 Population differences in proteome-defined pathway activity scores.** Shown are population differences in the abundance of proteins driving (**a**) autophagy, (**b**) apoptosis, (**c**) chemotaxis, (**d**) promotion of tumor immunity, (**e**) suppression of tumor immunity, and (**f**) vasculature. Heatmaps despict levels of process/pathway-associated proteins in relationship to population group (Ghanaian, AA, EA). Shown to the right are the mean score differences for these processes/pathways among the three population groups. Profiles for Ghanaian ($n = 654$), AA ($n = 374$), and EA ($n = 454$) men without prostate cancer are shown. The process/pathway scores are derived from the average z-scores of all the associated proteins. These scores are shown as violin plots and were compared using two-sided Wilcoxon rank sum tests. P values were adjusted for multiple comparison. FDR-adjusted P value significance was coded as <0.0001 (****), <0.001 (***), and ≥0.05 (ns). The violin plots represent median values ± IQR. Source data are provided as a Source Data file. TI tumor immunity. FDR False Discovery Rate, AA African American, and EA European American.

pleiotrophin (PTN), both positively associated with the risk of lethal disease, and regular aspirin use (negatively associated with risk) emerged as a top predictor based on two selection criteria: the feature frequency (Fig. 7a) and the weight of the features' contribution to the prediction (Fig. 7b). These three features combined predicted prostate cancer-specific mortality with an accuracy of 83.7% (SE = 3.8%). Our finding that regular aspirin

use was a predictor of improved survival among AA men is consistent with previously published data from this case-control study[14] and the Southern Community Cohort Study[15]. The combination of the two proteins alone predicted prostate cancer-specific mortality with 78.2% (SE = 4.2%) accuracy. To gain additional insight of how the prediction of lethal prostate cancer by these two protein markers might be influenced by a patient's

**a**

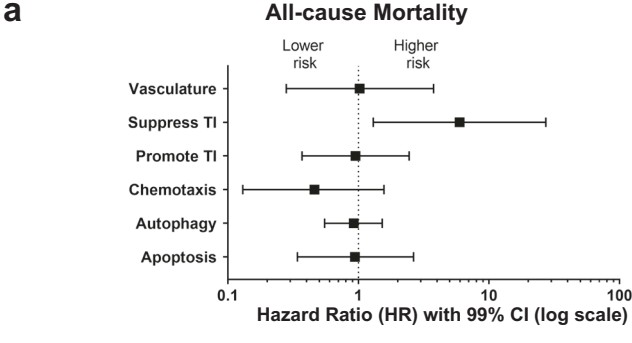

**b**

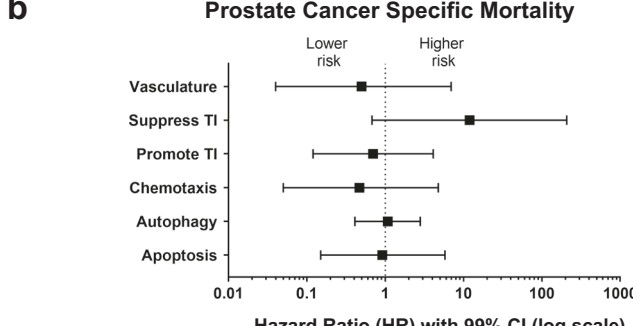

**c**

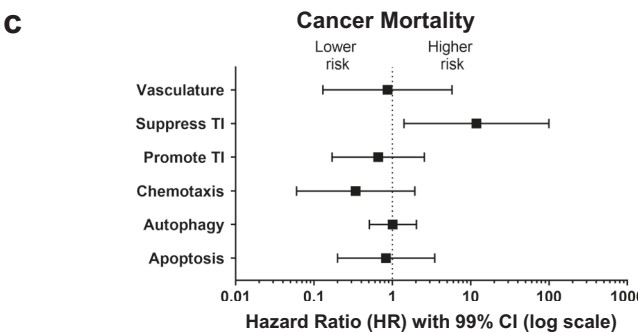

**Fig. 6 Suppression of the tumor immunity pathway associates with survival of prostate cancer patients.** We assessed the association of the six pathways defined by the 82 immune-oncology markers with all-cause mortality ($n = 202$), prostate cancer-specific mortality ($n = 57$), or mortality due to any cancer after a prostate cancer diagnosis ($n = 103$) out of the 819 prostate cancer patients followed. The pathway scores were evaluated as continuous predictor variables. Suppression of tumor immunity pathway was distinctively associated with all-cause mortality (**a**), prostate cancer-specific mortality (**b**), or a mortality due to any cancer after a prostate cancer diagnosis (**c**). Multivariable Cox regression analyses were used to assess if the pathways were independently associated with survival of prostate cancer patients in the NCI-Maryland study. For (**a**–**c**), the hazard ratios (HRs) were adjusted for age at study entry (years), BMI (kg/m2), self-reported race (AA/EA), education (high school or less, some college, college, professional school), income (<\$10k, \$10–30 K, \$30–60 K, \$60–90k, greater than \$90k), smoking history (never, former, current), diabetes (no/yes), aspirin use (no/yes), treatment (0 = none, 1 = surgery, 2 = radiotherapy, 3 = hormone, 4 = combination), and NCCN risk score. The HRs indicate the change in risk of dying when the biological process z-score value increases by 1 while holding all the other biological processes' z-scores and covariates constant. Data are presented as hazard ratios ± 99% confidence intervals. Source data are provided as a Source Data file. TI tumor immunity and CI confidence interval.

disease status, we added the NCCN risk score to our model. Consistent with the baseline model that did not contain the NCCN risk score, lasso (alpha = 1) remained the most predictive model (Supplementary Fig. 7b, Supplementary Data 5). The

NCCN risk score was the top predictor of the lethal disease. The two analytes PTN and TNFRSF9 remained the most predictive features besides the NCCN risk score, and the three features combined predicted prostate cancer-specific mortality with 90% accuracy (Supplementary Fig. 7B, Supplementary Fig. 8). In addition, AA prostate cancer patients with high levels (>median) of both TNFRSF9 and PTN in their blood at diagnosis had the worst prostate cancer-specific survival independent of disease status [(Adjusted HR = 3.09 (1.36, 7.03)] (Fig. 7c). By 10 years, 33% of cases with high levels of both TNFRSF9 and PTN died of prostate cancer compared to only 5% of cases with low levels of one or both of these proteins (Fig. 7c), highlighting the utility of these blood markers for risk stratification of AA prostate cancer patients.

## Discussion

In this study, we describe differences in the expression of immune and chemotaxis-related markers in men from three population groups, with two of them—AA and Ghanaian men—having an ancestral relationship due to the trans-Atlantic slave trade. Most notably, expression of immune-oncology markers related to immune suppression were up-regulated in men of West African ancestry and were associated with prostate cancer mortality. While ancestry can explain some of the observations, other and yet unknown factors contribute to these clinically significant differences in immune function and chemotaxis.

Infections endemic to certain regions have shaped the immune response in affected populations, leaving a lasting genetic and epigenetic footprint[33]. As such, population differences in exposures to fatal pathogens have led to population heterogeneity in the immunome. It has been estimated that as many as 360 immune-related genes have been targets of positive selection and have functional variations between populations[34]. Consistent with these observations, we now report population differences in circulating immune-oncological proteins among Ghanaian, AA, and EA men. We found that the serum proteome-defined immunome of Ghanaian men resembles the immunome of AA men more so than EA men. We identified CXCL5, CXCL1, MCP2, MCP1, and CXCL11 as the top immune-oncological proteins associated with West African ancestry. Four of these chemokines (CXCL5, CXCL1, MCP1, and CXCL11) are known targets of Duffy Antigen Receptor for Chemokines (DARC) binding[35]. DARC is a non-signaling receptor that binds to both CXC and CC family of chemokines and acts as a depot for chemokines on erythrocytes and as decoy receptor on endothelial cells[36]. DARC expression modulates the susceptibility to clinical *Plasmodium vivax* malaria and loss of its expression on erythrocytes, which frequently occurs in sub-Saharan African populations due to germline genetic variants, confers resistance against malarial infection[37]. Its loss may also influence cancer susceptibility[38,39]. Consequently, these individuals lack the ability to sequester the target chemokines, leading to elevated concentration of the chemokines in circulation[40]. Accordingly, we found that CXCL5, CXCL1, and CXCL11 were 2–3-fold higher in sera of Ghanaian or AA men than EA men. Given the angiogenic properties of these chemokines[41], their role in cancer progression has been proposed[42].

As a key finding, we report that serum proteins regulating chemotaxis and suppression of tumor immunity were elevated in men of African ancestry, suggesting persistent population differences in stimulation of leukocyte recruitment and T cell mediated immune response. Such differences may predispose men of African descent to a distinct tumor microenvironment. Although the direct impact of the peripheral immunome on the prostate tumor microenvironment requires further investigation,

**Table 1 A high score for suppression of tumor immunity associates with National Comprehensive Cancer Network (NCCN) Risk Score for metastatic prostate cancer.**

| NCCN risk score | Total OR (95% CI)[a] | AA OR (95% CI) | EA OR (95% CI) |
|---|---|---|---|
| Low | Ref. | Ref. | Ref. |
| Intermediate | 1.04 (0.68–1.59) | 0.89 (0.46–1.70) | 1.18 (0.65, 2.13) |
| High/Very High | 1.47 (0.87–2.48) | 1.33 (0.59–2.98) | 1.72 (0.83, 3.54) |
| Regional/Metastatic | **3.79 (1.59–9.04)** | **5.90 (1.43–24.34)** | 3.16 (0.95, 10.50) |
| *P* value for trend | **0.004** | **0.019** | **0.040** |

Bolded data indicate significant associations in the logistic regression analysis.
High suppression of tumor immunity is defined by the median score in the NCI-Maryland control population (>median vs. ≤median).
[a]Logistic regression adjusted for age at study entry, BMI (kg/m2), diabetes (no/yes), aspirin (no/yes), education (high school or less, some college, college, professional school), family history of prostate cancer (first-degree relatives, yes/no), self-reported race (not included in the stratified analysis), income (<$10k, $10–30 K, $30–60 K, $60–90k, >$90k), smoking history (never, former, current), treatment (0 = none, 1 = surgery, 2 = radiation, 3 = hormone, 4 = combination).

we and others have previously reported stark differences in the immune landscape of prostate tumors of AA men as compared to EA men[9–13]. For instance, programmed cell death ligand-1 (PD-L1), which suppresses T cell–mediated tumor immunity, was found to be overexpressed in AA prostate tumors[43]. Recent work by Awasthi et al. reported that AA prostate tumors tend to be enriched for immune pathways that are associated with poor clinical outcomes[44]. We show with our current work that elevated, peripheral suppression of tumor immunity associates with lethal prostate cancer and the underlying mechanism may possibly involve an effect on metastasis. Hence, population differences in suppression of tumor immunity may contribute to the disproportionate burden of lethal prostate cancer among men of African ancestry. On the other hand, such differences may offer a therapeutic advantage for immunotherapeutic strategies that are tailored to target immune suppressive pathways. A recent study provided a first indication that differences in the response to cancer vaccines may lead to higher survival rates among AA men[45].

Differentiating men who have lethal forms of prostate cancer from those with a more slow-growing disease remains a major challenge in clinical oncology. Risk stratification strategies are particularly needed for AA prostate cancer patients who disproportionately bear the prostate cancer burden. This study identified TNFRSF9/CD137/4-1BB and PTN as candidate predictive blood markers for prostate cancer mortality among AA patients. AA patients with high levels of both TNFRSF9 and PTN in their sera had the highest risk of dying from prostate cancer. The membrane form of TNFRSF9 possesses antitumor properties and agonistic anti-TNFRSF9 antibodies are currently in clinical trials[46,47]. On the contrary, the soluble isoform of TNFRSF9 that we measured, generated by alternative splicing[48], has been shown to antagonize antitumor immune response hence promote tumor survival most likely by acting as decoy receptor[49,50]. Regulatory T cells described as Tregs are thought to be a major source of secreted TNFRSF9[51,52]. Recently, TNFRSF9 mRNA level was shown to be a robust marker of tumor-infiltrating Tregs that suppress antitumor response[53]. Moreover, high numbers of TNFRSF9–expressing Tregs were associated with poor survival outcomes across multiple human cancers[53], consistent with our findings that serum TNFRSF9 associates with lethal prostate cancer. PTN or pleiotrophin, the second protein marker found to be associated with lethal prostate cancer in AA men, may not have the same immune function that soluble TNFRSF9 exhibits. PTN is a secreted cytokine that is developmentally regulated. Normally expressed during embryogenesis as a growth or differentiation factor, it is expressed either at very low levels or not at all in healthy adults[54,55]. PTN re-expression in adults is associated with tumor development, metastasis and angiogenesis with elevated expression reported in several cancer sites[56–58]. This

increased expression of PTN has been associated with poor prognosis in colorectal cancer[56], hepatocellular carcinoma (HCC)[58] and gliomas[57]. Several hypotheses have been pursued to find out how PTN exerts pro-metastatic effects. For example, it is proposed that PTN promotes cancer progression through increased vascular endothelial growth factor deposition at the vasculature leading to vascular disruption[57]. PTN may upregulate lipid synthesis, contributing to hepatic stenosis and progression of HCC[58]. Serum PTN is a candidate biomarker for occurrence of breast[59] and small cell lung cancers[60] and has been shown to associate with metastatic prostate cancer[61], consistent with the findings in this study. In prostate cancer, PTN regulates mesenchymal and epithelial proliferation, with PTN itself being regulated by the androgen receptor during prostate development[62]. To the best of our knowledge, a potential relationship between PTN and TNFRSF9 has not been described. However, PTN has been implicated in the induction of TNF-α expression in peripheral blood mononuclear cells demonstrating a link between PTN and the TNF superfamily in the circulation[63].

The immune and inflammatory environment in the circulation has been implicated as a potential influencer of metastasis. Evidence is emerging to indicate that inflammation-activated platelets are pro-metastatic, instigating the formation of a pre-metastatic niche. *Lucotti et al.* recently demonstrated reduced metastasis in lung cancer models through inhibition of intra-vascular COX-1-derived thromboxane A2 (TXA2) from platelets via aspirin treatment[64]. Consistent with the animal model data, we recently reported an association between elevated urinary TXB2 (the stable metabolite of TXA2) and metastatic prostate cancer in AA men with prostate cancer[65], suggesting a distinct inflammatory environment and platelet activity in these men. Platelets can disrupt immunosurveillance of the metastatic cascade through cloaking of natural killer (NK) cells, preventing the NK cells from patrolling and inducing tumor-cell cytolysis[66]. Pre-clinical studies have also implicated other immune cells including macrophages[67], Tregs[68], and neutrophils[69,70] as promoters of the metastatic process through protection of tumor cells in the circulation and promotion of tumor cell seeding. Thus, the immune-oncological profile in men of African ancestry may develop in an environment of systemic chronic inflammation and promote metastasis. Future research should test this hypothesis.

Our study has strength and limitations. The major strength is the large sample size, the measurement of 82 immune-oncology markers with a robust technology, and the inclusion of men from Ghana and the U.S. Moreover, we applied multiple testing adjustments in reporting the significance of our observations. However, we collected blood samples in Ghana and the U.S. Although blood sample collection in Ghana followed a protocol

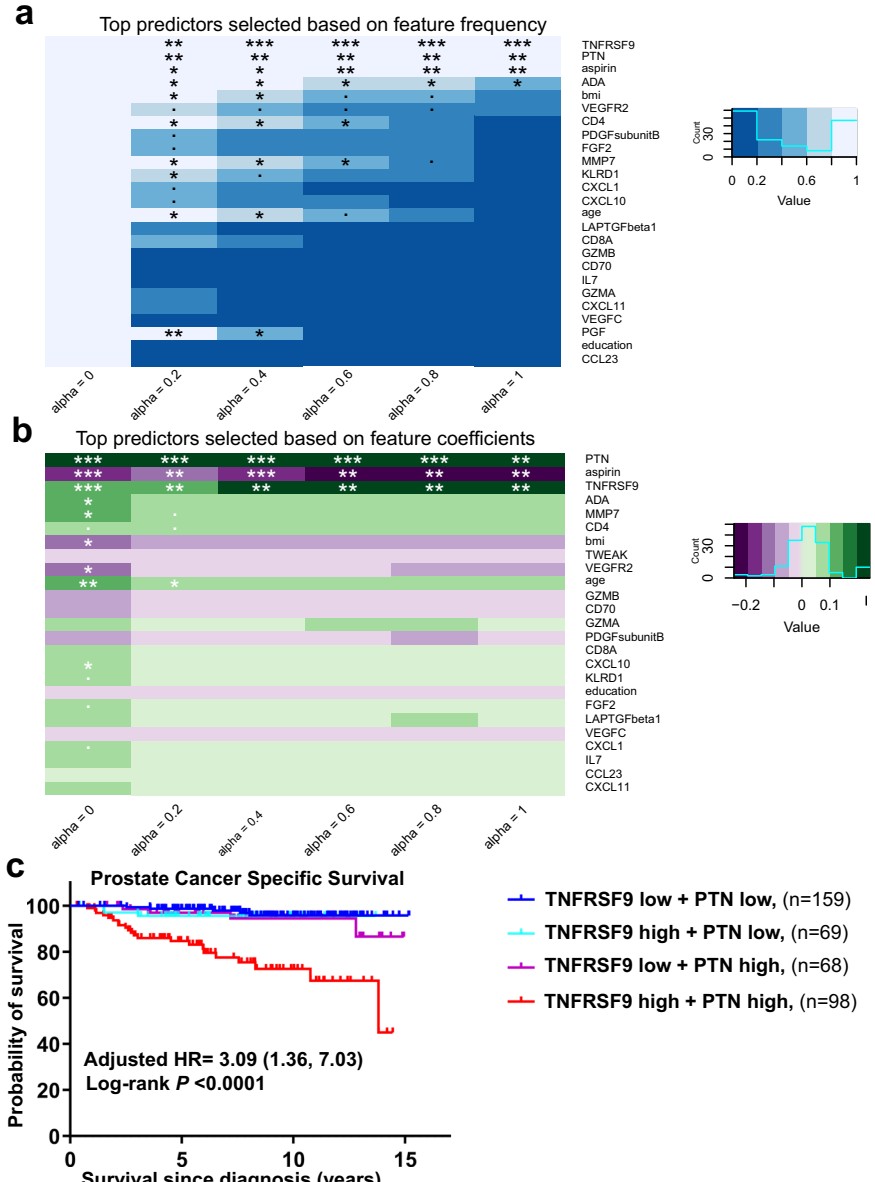

**Fig. 7 A signature of two serum markers is predictive of lethal prostate cancer in AA patients.** Cross-validated, regularized Cox regression models with different elastic net mixture parameters from ridge (alpha = 0) to lasso (alpha = 1) were implemented to identify a predictive proteomic signature. One-sided *P* values were obtained empirically by comparing feature frequencies and coefficients against those obtained from random permutations of the response[74]. **a** Heatmaps of feature frequencies across alpha. Features were ranked by *P* value for alpha = 1. **b** Heatmaps of feature coefficients across alpha. Features were ranked by *P* value for alpha = 1. **c** Kaplan–Meier plot comparing prostate cancer-specific mortality of AA cases with high levels (>median) of both TNFRSF9/CD137/4-1BB and pleiotrophin (PTN) vs. low levels of either or both proteins. Two-sided log-rank test was used to determine if there were statistically significant survival differences. Adjusted hazard ratio (HR) compares the risk of prostate cancer mortality for those with high levels of both TNFRSF9 and PTN vs. the remaining AA cases. HR estimates were adjusted for potential confounding factors: age, BMI, education, income, smoking history, diabetes status, aspirin use, treatment, and NCCN risk score. In (**a** and **b**), *P* value significance was coded as <0.001 (***), <0.01 (**), <0.05 (*), and <0.1 (.). The exact *P* values are found in the Source Data file. Source data are provided as a Source Data file. HR hazard ratio.

that applied standards of practice in the U.S., we cannot exclude that serum preparation and shipping influenced the performance of the immune-oncology marker measurements, yet we are not aware of such an influence. In our survival analysis, we included 57 events for prostate cancer-specific deaths, 103 events for any cancer deaths, and 202 events for all-cause mortality among men with prostate cancer. In these analyses, we found that an elevated suppression of tumor immunity score to beconsistently associated with decreased survival. Yet, additional studies will be needed to further confirm the association of the suppression of tumor immunity signature with lethal prostate cancer.

In conclusion, it is a key finding of our study that suppression of tumor immunity is increased in Ghanaian and AA men, when compared to EA men, and associates with lethal prostate cancer. As such, these findings provide an insight into potential causes of the prostate cancer health disparity. The current study has a large representation of men of African descent who were profiled for their immune-oncological proteome. With the advent of an increasing number of immunotherapies in the drug development pipeline, studies like ours may inform clinical research on population differences in the immune landscape that need to be considered when designing therapies that exploit the immune response.

## Methods

**NCI-Maryland prostate cancer case-control study.** This study and the eligibility criteria have been previously described[14,71]. Race/ethnicity was assigned based on self-identification as either black or AA or as white or EA. The study was initiated to test the primary hypothesis that environmental exposures and ancestry-related factors contribute to the excessive prostate cancer burden among AA men. The study was approved by the NCI (protocol # 05-C-N021) and the University of Maryland (protocol #0298229) Institutional Review Boards and all participants signed an informed consent. Cases were recruited at the Baltimore Veterans Affairs Medical Center and the University of Maryland Medical Center. A total of 976 cases (489 AA and 487 EA men) were recruited into this study between 2005 and 2015. Controls were identified through the Maryland Department of Motor Vehicle Administration database and were frequency-matched to cases on age and race. A total of 1034 population controls were recruited (486 AA and 548 EA men). At the time of enrollment, both cases and controls were administered a survey by a trained interviewer and a blood sample was collected. Serum samples were available for 846 cases (407 AA and 439 EA) and 846 controls (382 AA and 464 EA), therefore only these individuals were used for the study herein. Most of the 846 cases (85%) were recruited within a year of the disease diagnosis with a median of 5.1 months between disease diagnosis and blood collection.

**NCI-Ghana prostate cancer case-control study.** This case-control study has been previously described[20]. The study was designed to study lifestyle, environmental, and genetic risk factors for prostate cancer in African men. The study was approved by institutional review boards at the University of Ghana (protocol #001/01-02) and at the National Cancer Institute (protocol #02CN240). Prior to study enrollment, all participants signed an informed consent. Prostate cancer cases were recruited at Korle Bu Teaching Hospital in Accra, Ghana between 2008 and 2012. The cases were diagnosed using Digital Rectal Exam (DRE) and PSA tests, followed by biopsy confirmation. Immediately after diagnosis and before treatment, cases were consented and asked to submit blood specimen and questionnaire data. Controls were identified through probability sampling using the 2000 Ghana Population and Housing Census data to recruit ~1000 men aged 50–74 years in the Greater Accra region between 2004 and 2006. These men were confirmed to not have prostate cancer by PSA testing and DRE. Serum samples were available for 586 prostate cancer cases and 659 population controls; hence, only these individuals were used for the study herein.

**Serum sample processing.** The participants in the two studies provided blood samples at time of recruitment. For the NCI-Maryland study, most blood samples were processed the same day, but always within 48 h, after storage in a refrigerator. For the NCI-Ghana study, blood samples were processed within 6 h. In this study, population controls provided fasting blood. Serum was prepared using standard procedures and aliquots were stored at −80 °C. Serum samples were shipped from Ghana to the NCI in dry ice boxes.

**Serum protein measurement.** Serum levels of 92 immuno-oncology panel proteins were measured simultaneously using a proprietary multiplex Proximal Extension Assay by Olink Proteomics (Boston). Olink utilizes a relative quantification unit, Normalized Protein eXpression (NPX), which is in a Log2-format. Serum samples from NCI-MD study (846 cases and 846 controls) and NCI-Ghana study (586 cases and 659 controls) were completely randomized and were assayed in that order. In addition to the built-in internal controls, 5% blinded duplicates were randomly selected and were randomized along with the original set of samples. Protein levels were intensity normalized to adjust for batch effect. Because all our samples were randomized across plates, a global adjustment was used to center the values for each assay around its median and across all plates. Ninety-five percent of the samples passed a stringent quality control (1647 from the NCI-MD study: 819 cases and 828 controls; 1143 from NCI-Ghana study: 489 cases and 654 controls)—with coefficients of variation among duplicates at <10% for every marker. These are men reported in Supplementary Table 1. Out of the 92 proteins assayed, IL33, IL35, IL21, IL2, IFNβ, IL13, IL1α, CXCL12, IFNγ, and TNF were detected in <20% of the samples, hence the remaining 82 proteins were used for subsequent analysis (Supplementary Table 3).

**Functional annotation and biological processes scores.** Proteins were grouped into six biological processes based on their respective biological roles following the Olink guideline: apoptosis/cell killing, autophagy/metabolism, chemotaxis/trafficking to tumor, suppression of tumor immunity (Th2 response, tolerogenic), promotion of tumor immunity (Th1 responses), or vasculature and tissue remodeling. Apoptosis, autophagy, chemotaxis, suppression of tumor immunity, promotion of tumor immunity, or vasculature scores were calculated for each study participant as the mean z-score value for the proteins belonging to the respective biological process. For survival analysis, the biological process/pathway scores were evaluated as continuous variables. To evaluate the association of suppression of tumor immunity with aggressive prostate cancer, we grouped suppression of tumor immunity scores into low (≤median) and high (>median) with cutoffs determined using the distribution of the score among population controls of the NCI-Maryland study.

**Prostate specific antigen (PSA) measurement.** For the cases in the NCI-Maryland cohort, PSA levels were obtained from medical record. For the controls of the NCI-Maryland study, total PSA was measured from stored serum aliquots using the human total PSA ELISA Kit (Abcam, ab188388). About 7% ($n = 56$) of the controls in the NCI-Maryland cohort had PSA >2.5 ng/ml, while only 3% ($n = 27$) had blood PSA over 4 ng/ml. For the controls in the NCI-Ghana study, close to 20% ($n = 132$) had a PSA >2.5 ng/ml, while about 11% ($n = 73$) had PSA over 4 ng/ml.

**C-reactive protein (CRP) measurement.** Plasma CRP was assayed using an ELISA assay (cat# ab99995, Abcam, United States) according to the manufacturer's instructions. Two microliters of plasma samples were added to 398 μL of 1x Diluent D, followed by a second 1:200 dilution steps for each sample. One-hundred microliters of CRP standard (0–600 pg/mL) and the diluted samples were loaded as duplicates into pre-coated 96-well plates. Samples were incubated overnight at 4 °C with gentle shaking, followed by incubations with the anti-human CRP antibody and the horseradish peroxidase-streptavidin solution. CRP was quantified measuring absorbance at 450 nm with a microplate reader.

**West African ancestry estimation for participants in the NCI-Maryland case-control study.** Genomic DNA was isolated from buffy coats (DNeasy Blood & Tissue Kit - Qiagen) or mouthwash samples (standard phenol-chloroform technique). Isolated DNA was genotyped for 100 ancestry informative markers using the Sequenom MassARRAY iPLEX platform, as previously described[30]. Single nucleotide polymorphism (SNP) genotype calls were generated using Sequenom TYPER software for 1505 of the 1647 (91%) individuals from the NCI-Maryland prostate cancer study with QCed serum proteomics data (i.e., 710 cases and 795 controls). A genotype concordance rate of >99% was observed for all markers. Admixture estimates for each study participant were calculated using a model-based clustering method as implemented in the program STRUCTURE v2.3. We applied STRUCTURE v2.3 with an admixture model estimating K (number of sub populations) from 2 to 5 with 100 iterations and parental population genotypes from West Africans, Europeans, and Native Americans, yielding three admixture estimations (West African, European, Native American). For a subset (83%) of the NCI-Maryland study participants with QCed serum proteomics data (i.e., for 1367 men: 658 cases and 709 controls), additional West African ancestry estimates were provided by the Cancer Genomics Research Laboratory/NCI-Leidos from a genome-wide association study using the Infinium HumanOmni5-Quad BeadChip array. Here, we employed the SNP weights approach for ancestry estimation which is a python-based software for ancestry inference using genome-wide SNP weights precomputed from external reference panels[72]. We interrogated a total of 55,446 genotype SNPs after applying linkage disequilibrium-based pruning and minor allele frequency filtering and those served as input SNPs. SNP weights will check for reference allele and strand between the pre-computed SNP weights and input genotypes. It then uses the intersection of the SNP weights and input genotypes to perform ancestry inference. The West African ancestry estimates using the two approaches were similar ($r = 0.98$) and can be found in Supplementary Data 6.

**Association of clinical/socio-demographic characteristics with immune-oncological proteins.** The association of age, BMI, education, aspirin use, smoking, diabetes, and PSA levels with the relative abundance of individual analytes (as continuous value) was assessed by means of multivariable linear regression models implemented by the function lm in the base R package stats (version 3.6.1). These variables were chosen because they have either been linked to prostate cancer risk and survival or may influence the status of inflammation and host immunity. For each analyte, we fitted the formula "analyte ~ age + bmi + education + aspirin + smoking + diabetes + PSA", which yielded the model's F-statistic and associated F-statistic $p$ value, as well as the intercept and regression coefficients with their associated standard errors (SE) and $P$ values. F-statistic $P$ values were adjusted by FDR across all models; moreover, within each model, regression coefficient $P$ values were also FDR-adjusted. Full regression results for each cohort are provided as Supplementary Data 1. In Fig. 2, an analyte (as response variable) was considered significantly associated with clinical and socio-demographic covariables if the multivariable model yielded an FDR-adjusted $P$ value < 0.05 on the F-statistic. If this condition was satisfied, the association between the target analyte and each individual covariable was characterized by the corresponding FDR-adjusted $P$ value and coefficient. In order to assess the statistical significance of cohort differences in the association between analytes and clinical/socio-demographic characteristics, we fitted each analyte to a model with interaction terms according to the formula "analyte ~ age*Gh + bmi*Gh + education*Gh + aspirin*Gh + smoking*Gh + diabetes*Gh + PSA*Gh + age*Afr + bmi*Afr + education*Afr + aspirin*Afr + smoking*Afr + diabetes*Afr + PSA*Afr", where the intercept and non-interacting terms were implicitly also included in the model. Here, two dummy variables were introduced: Gh (defined as 1 for Ghanaian subjects, 0 otherwise) and Afr (defined as 1 for Ghanaian and African American subjects, 0 for European Americans). As described above, models were selected based on significance of the FDR-adjusted $p$ value < 0.05 on the F-statistic; if this condition was satisfied, the association between the target analyte and each individual covariable was characterized by the corresponding FDR-adjusted $P$ value and coefficient.

**Analysis of variance as a function of genetic estimation of West African admixture**. Variance analysis for the levels of each of the 82 immune-oncological cytokines were simultaneously assessed as a function of genetic estimation of West African admixture among men without prostate cancer from the NCI-Maryland study. The analysis was implemented by the function aov in the base R package stats (version 3.6.1).

**Analysis of (dis)similarity across cohorts**. Using the distance metric $d = 1-r$, where $r$ is Pearson's correlation between pairs of subjects, unsupervised hierarchical clustering was performed using average linkage and visualized as a heatmap with cohort annotations (Fig. 3), generated using Broad Institute's web-based matrix visualization and analysis platform Morpheus (https://software.broadinstitute.org/morpheus). To avoid spurious effects from outliers in heatmap plots, each protein's range of abundance values were set to saturate at the 1st and 99th percentiles. To account for widely different abundance ranges for different proteins in the assay, each protein's measured abundances across all subjects were z-score transformed. The hierarchical clustering dendrogram was cut to extract K clusters (with K = 2, 3). The association between cluster labels and population groups was tested via Fisher's or chi-squared tests performed on the resulting contingency tables (Supplementary Fig. 5).

**Gene ontology (GO) enrichment analysis**. GO terms with an enrichment in proteins of interest were identified using Over-Representation Analysis (ORA) as part of the web tool WebGestalt (WEB-based Gene SeT AnaLysis Toolkit). Enriched gene sets were further processed using affinity propagation (R package apcluster) to cluster gene sets according to functional similarity.

**Survival analysis**. Information on patient survival was only obtainable for the NCI-Maryland prostate cancer patients. Survival data was obtained from the National Death Index database for both cases and controls in the NCI-Maryland study. We calculated survival for cases from date of diagnosis to either date of death or to the censor date of December 31, 2018. We built a multivariable Cox regression model with all biological processes scores and adjustment for other covariates to estimate adjusted HRs and 95% or 99% CIs for all-cause mortality, cancer-related mortality, and prostate cancer-specific mortality. We adjusted for the following potential confounding factors: age at study entry (years), BMI (kg/m$^2$), self-reported race (AA/EA), education (high school or less, some college, college, professional school), income (>$10k, $10–30 K, $30–60 K, $60–90k, > $90k), smoking history (never, former, current), diabetes (no/yes), aspirin use (no/yes), treatment (0 = none, 1 = surgery, 2 = radiotherapy, 3 = hormone, 4 = combination), and disease status defined by the NCCN risk score. Missing values for education ($n = 1$), smoking history ($n = 5$), and income ($n = 63$) were imputed using the R package missForest, which implements nonparametric missing value imputation based on random forests. In the overall survival analysis of population controls, we calculated survival from the date of interview to either date of death or to the censor date of December 31st, 2018. We applied the Cox regression model to estimate adjusted HR and 95% CI and adjusted for all the confounding factors listed above except for treatment. Missing values for education ($n = 1$), smoking history ($n = 7$), and income ($n = 67$) were imputed using the R package missForest. The reported HRs indicate the change in risk of dying when the biological process z-score value increases by 1 while holding all the other biological processes' z-scores and covariates constant.

**Classification of cases using National Comprehensive Cancer Network (NCCN) risk score**. Cases were assigned to risk groups based on the patients' TNM stage, Gleason score, Gleason pattern, and PSA level at diagnosis according to the 2019 NCCN guideline for prostate cancer[73]. Information on TNM stage was only obtainable for the NCI-Maryland prostate cancer patients, hence only these cases were scored. Cases were categorized as localized, regional, and metastatic prostate cancer based on their clinical parameters at the time of diagnosis. Localized prostate cancer cases were further classified into low, intermediate, high, and very high risk based on the likelihood of their disease to progress to lethal prostate cancer per the 2019 NCCN guideline[73]. Prostate cancer cases with lymph node involvement but no distant metastasis at diagnosis were classified as regional prostate cancer while those with distant metastasis at the time of diagnosis were classified as metastatic prostate cancer. For our analysis, we condensed these risk groups into four categories (low, intermediate, high/very high, and regional/metastatic).

**Developing a predictive proteomic signature of lethal prostate cancer**. The analysis was restricted to the cases from NCI-Maryland study for whom we had survival data. We stratified by self-reported race/ethnicity into AA cases (360 censored, 34 prostate cancer deaths) and EA cases (402 censored, 23 prostate cancer deaths). To identify a multi-analyte proteomic signature that is predictive of lethal prostate cancer, 88 features were evaluated [82 immune-oncological proteins along with six demographic/clinical variables (education, age, BMI (kg/m$^2$), smoking history, diabetes, and aspirin use)]. Missing values for education ($n = 1$) and smoking history ($n = 5$) were imputed using R package missForest. R package eNetXplorer (version 1.1.2)[74] was implemented to build cross-validated,

regularized Cox regression models with different elastic net mixture parameters from ridge (alpha = 0) to lasso (alpha = 1). Alpha was selected based on overall performance assessed as a function of the fivefold cross-validated quality function (concordance) and the empirical $P$ value generated from comparing the model against a statistical ensemble of null models created by random permutations of the response (i.e., survival time/status randomized across subjects in the cohort). These results comprise 10,000 Cox regression elastic net realizations arising from 200 randomly generated folds, each of them compared against 50 null model permutations. Features' performance as predictors was evaluated using two different, but complementary selection criteria: feature coefficients and feature frequencies. The feature frequency measure captures the significance of how often a feature is chosen in an in-bag model. When it is chosen, the feature coefficient measure captures the significance of the feature's weight in the in-bag model. See the publication by *Candia et al.* for more details on this method[74]. Using only the significant protein features from both selection criteria, a multivariate Cox regression model was run. Risk stratification was used to generate Kaplan–Meier plots and log-rank tests of significance.

**Statistical analysis**. Data analyses were performed using Stata/SE 16.0 and R statistical packages. An association was considered statistically significant with $P < 0.05$ or FDR-adjusted $P < 0.05$ in instances where correction for multiple testing was required.

**Reporting summary**. Further information on research design is available in the Nature Research Reporting Summary linked to this article.

## Data availability

Clinical, demographic and molecular data used for this study (i.e., self-reported race, degree of West African ancestry, age, BMI, education, income, aspirin use, diabetes use, smoking status, NCCN risk score, PSA, treatment type, proteomics data, and survival data) are deposited at the Open Science Framework (https://osf.io/327ha (https://doi.org/10.17605/OSF.IO/327HA)[75] and as a public GitHub repository at https://github.com/juliancandia/ProstateCancerProteomics (https://doi.org/10.5281/zenodo.5815262)[76]. The individual West African ancestry estimates for participants in the NCI-Maryland study obtained with either 100 ancestry informative markers or 55446 GWAS-based SNPs can be found in Supplementary Data 6. Individual raw genotype data cannot be shared through open access due to NIH rules that do not allow it because the participants did not consent to sharing this type of data. Source data are provided with this paper. The remaining data are available within the paper, Supplementary Information, and Supplementary Data. Source data are provided with this paper.

## Code availability

The scripts used in our bioinformatics pipeline to perform data analysis and visualization are available as a public GitHub repository at https://github.com/juliancandia/ProstateCancerProteomics (https://doi.org/10.5281/zenodo.5815262)[76].

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

## Acknowledgements

We would like to thank personnel at the University of Maryland and the Baltimore Veterans Administration Hospital for their contributions with the recruitment of participants to the NCI-Maryland study. We would also like to thank Prof. Edward D. Yeboah as the original Ghana PI and Ms. Evelyn Tay as the original Study Manager for the NCI-Ghana study. This work was supported by the following grants: DoD award W81XWH1810588 (to S.A., C.Y.), U54 CA118623- CY (NCI) and U54-MD007585-26-CY (NIMHD) (to C.Y.), and Intramural Research Program of the NIH, National Cancer Institute (NCI), Center for Cancer Research and Division of Cancer Epidemiology and Genetics (to S.A., M.B.C.).

## Author contributions

Conceptualization: T.Z.M., C.Y., M.B.C., S.A. Data curation: T.Z.M., T.H.D., M.K., C.J.S., S.V.J., A.L.Z., O.M.O., A.A., F.J.J., R.K. Formal Analysis: T.Z.M., J.C., R.K. Funding acquisition: C.Y., M.B.C., S.A. Investigation: T.Z.M., J.C., F.J.J. Methodology: T.Z.M., J.C., C.A.L., M.B.C., S.A. Project administration: T.H.D., F.B. Resources: W.T., Y.T., R.B.B., A.A.A., J.E.M., R.N.H., A.W.H., M.B.C., S.A. Supervision: W.T., S.A. Visualization: T.Z.M., J.C. Writing—original draft: T.Z.M. Writing—review & editing: T.Z.M., J.C., F.B., W.T., M.K., C.J.S., Y.T., R.B.B., A.A.A., J.E.M., X.W.W., C.A.L., C.Y., M.B.C., S.A.

## Funding

## Competing interests

The authors declare no competing interests.
