## [Peer Review File · Nature Communications]

Reviewers' Comments:

Reviewer #1:

Remarks to the Author:

Minas et al

Serum Proteomics in AA vs A vs EA

For these studies, Minas et al performed obtained sera from men with prostate cancer who were African (Ghana cohort), African-American or European American. Serum proteomics were evaluated using the Olink technology, and a robust number of cases and controls were analyzed. The demographic of the pts from whom the samples came are described in the Materials and Methods section. The data are descriptive and a bit diffuse, but very interesting. Briefly they found that common demographic factors (age, BMI etc) associated differently with markers in different populations, that samples from men from Ghana clustered differentially from AA or EA samples, and they also validated some of these associations by quantifying genetic inheritance using genomic markers. That's very rarely done, and is a clear high point of the data. A signature associated with suppression of tumor immunity was associated with ACM and CM across pts. Finally, a reductionist two gene signature including only 41BB (TNFRSF9 and PTN) was strongly associated with PC mortality but only in AA pts. Although the results are largely descriptive, the number of patients, robust statistical analyses and novel findings make this an important addition to the literature, and as such I have only minor comments.

1) Some key data, i.e. the association between different proteomic markers is buried in a supplemental Figure and table (S3 and table S3). I would strongly advocate for including these data in the main manuscript. Only a handful of studies have shown how cytokines associate, and those data are mostly from Luminex or mesoscale data. To my knowledge, this is probably the first such proteomic analysis performed, or at least the first analysis in PC pts, and as such is probably worthy of a little more attention.

2) Results – line 97: The phrase “were detected” needs a bit more clarification in the results section. Does each analyte have its own lower limit of detection? What are they? If available, they could be listed in a supplemental table. Similarly, for figure 1 it wasn't clear to me whether data were being correlated as continuous variables, or in terms of present vs absent. This should be clarified in the results section.

3) Results – line 88: ‘Patient characteristics have been previously described’. The authors absolutely need to expound a bit on patient characteristics here, it is a bit presumptuous to assume that the reader is expected to go back and read all three prior papers just to understand this one. Should consider saying “Briefly, the NCI-Ghana study included xx men with yy disease with a median age of xx. The NCI-Maryland PCCC included xx ... etc. Just 3 additional sentences should do the trick.

4) Results line 137- these extensive comments on CAIX are out of place here, they belong in the discussion section.

5) Figure 2 – unsupervised clustering. This figure doesn't add much, other than to demonstrate that the Af pts cluster differentially as compared to the AA and EA pts. The row text is too small to read, and the AA and EA are quite interspersed. My suggestion would be to move to move this figure to supplemental data if that were required to include supplemental table 3a and figure 3 as above.

6) Figure 6 – These are key data. Would suggest including mention that TNFRSF9 is CD137 / 4-1BB in the legend for clarity. PTN also needs to be defined in the figure legend as pleiotrophin, my guess is at least some readers are going to be a bit confused as to whether there's some serum marker of PTEN signaling or loss?

7) Discussion – The discussion on TNFRSF9 section is well thought out. Here CD137 / 4-1BB should also be mentioned for reference. The discussion of pleiotrophin is far too short; can the authors please provide some additional information on the function and expression pattern of this

analyte? They should also present at least one or two hypotheses on why PTN is associated with risk, and why it might co-associate with 41BB.

Reviewer #2:

Remarks to the Author:

Minas and colleagues have undertaken a remarkable dataset generation, creating serum proteomes on several thousand men, approximately equally divided between prostate cancer patients and controls. The dataset QC is extremely well-performed, and the resource will have very substantial value in and of itself.

1. Data Access

This dataset provides a major community resource. The authors have not indicated how all molecular and associated clinical and demographic data are available. That is central component of the value of this study.

2. Statistical Modeling

2a) The authors repeatedly incorrectly compare two features based on p-values within a single or different models. For example in lines 123-129 the authors incorrectly comparing models based on their p-values relative to the null. The correct approach is to compare the models directly, and state whether the coefficient for a gene is significantly different between the two cohorts: each model gives an effect-size (the coefficient) and its confidence interval. These need to be compared. Another example is lines 255-257 which compares AA to EA men. All statements of "exclusivity" of association to specific subgroups throughout need to be corrected. The two classic statistical approaches are directly comparing coefficients and their confidence-intervals (or other variance measure) as noted above, or by fitting a single multivariate model with stratification or other control and extracting the comparison of interest directly using a contrast matrix.

2b) Multiple testing is repeatedly ignored. For example, Figure 1 needs to be adjusted for multiple-hypothesis testing, as do many other areas (e.g. Table S5, Figure 4, etc.). The authors sometimes note a correction, but also present unadjusted results. The lack of correction is sometimes fatal to conclusions being drawn. For example the authors highlight in Figure 5B that elevated suppression of tumor immunity is associated with PCSM. Unfortunately this is marginally significant ($p = 0.057$) and the result is not adjusted for multiple testing. In fact the majority of survival analyses appear not to survive multiple testing correction. Multiple-testing correction is not optional: it must be applied throughout, and systematically in a uniform way. And only adjusted results should be presented.

2c) The methods are unclear: was a separate model fit for each clinical/socio-demographic variable (with each protein), or was a single model fit for each protein with all variables included? If separate models were fit, then the authors need to update to a single model. If a single model was fit, then the methods need rewriting for clarity (presumably with an equation).

2d) Line 180-182, looking at dendrogram groupings is not an appropriate way of determining proximity of groups. A formalized statistical analysis is required using distance metrics. Further, a separate hypothesis is that there is a cohort-wise difference in sample collection, storage, extraction or other handling. Have the authors excluded this possibility? If so, please outline more clearly in the text. If not this is a significant caveat to several of the analyses that needs to be clearly noted.

2e) The fraction of variance explained by ancestry (Table S6) seems largely uninterpretable based on p-values. Because of the large sample-size, even tiny effect-sizes are statistically significant. For example, the authors are indicating that 0.6% of variance explained by ancestry is a statistically significant result. Statistically, yes, but the biological interpretability of that tiny fraction is not clear. Of the 45 statistically-significant hits, only 7 explain even 10% in Table S6. Indicating p-values as a bright-line threshold independent of effect-size context is very poor statistical practice. Further, confidence intervals on these variance-explained estimates are required to interpret them appropriately.

2f) In a similar way, the authors report Spearman's rho for chemotaxis and tumor immunity suppression (line 218). All analyses need to be from appropriate multivariate models, reporting effect-sizes (not simply correlations) and their confidence-intervals.

2g) Survival models are intriguing, however the largest confounding factor is the state of the disease at diagnosis. The models do not appear to control for pre-treatment serum PSA, T-category, M-category, N-category or grade. The authors corroborate this concern by showing in the NCI-Maryland cohort that this score is preferentially associated with metastatic disease at diagnosis. So the conclusion appears to be "metastatic prostate cancer appears to occur in patients with enrichment of this functional group". The survival analyses are simply a distraction from uncontrolled models that confuse this analysis.

2h) Line 259 suggests that the authors aim to identify causal drivers, however none of the subsequent analyses indicate causality. The model presented in Figure 6 needs control for established clinical features associated with mortality (PSA, T, N, M, grade).

3. Writing

The discussion contains substantial speculation, and would benefit from trimming that and replacing with some detailed comments on the caveats of the study and analyses. The tone and wording of the study overall is too aggressive given the data at hand, and needs softening.

Reviewer #3:

Remarks to the Author:

The authors have written a very nice paper in which they describe different findings in three different populations (Ghanaian, African American (AA) and European American men) and measured 82 circulating proteins in almost 3000 men with and without prostate cancer. In men of West African ancestry protein signature for immunosuppression was significantly elevated. Two markers, pleiotrophin and TNFRSF9 predicted poor survival in AA men. I am very enthusiastic about this paper and it contains novel information.

As a clinician with a special interest in immunotherapy, I consider the paper well written with an appropriate interpretation of the immunological data.

I only have minor comments:

1. the initial PSA in the Af population is 52, whereas the iPSA in the other two groups (AA EA) is 6.8 and 6.0. It is probably due to later detection of prostate cancer in Ghana. However, more advanced disease could be associated with a more immunosuppressive environment. Please discuss that your findings are linked to ancestry and not to more advanced disease.
2. line 47, line 239, and line 535: You state in the paragraph about the statistical analysis that an association is only considered statistically significant if the P-value is less than 0.05. In your abstract and line 239 you are using phrasing like pointing to clinical significance and marginally significant. Please use the proper terms for significant and non-significant. I regret that in such a beautiful paper this misnomer is applied.

Reviewer #4:

Remarks to the Author:

Given evidence that tumor immunobiology and immunotherapy response differs in the treatment of African American versus European American men with prostate cancer, the authors determined if men with prostate cancer who are of West African ancestry inherit a unique immune-oncological signature pre-disposing them to greater health disparities compared to their non-African ancestry counterparts.

Noteworthy results of this study include 12/82 circulating immune-oncological proteins were significantly elevated in Ghanaian and African American men compared to European American

men. Specifically, TNFRSF9 and PTN predicted prostate cancer mortality with approximately 78% accuracy. The authors conclude that these findings not only contribute to elucidating the reasons for prostate cancer disparities in African American men but also inform future drug development research of putative prostate cancer immunotherapies.

This work contributes to the study of prostate cancer health disparities in African Americans because of its novel approach to identify a specific immune-oncological signature in African Americans that may provide more insight into prostate cancer health disparities in this group as well as assist in the development of more targeted therapies for this population which bears the burden of greater morbidity and mortality of prostate cancer.

I think the work supports the conclusions. However, I believe the authors should devote more effort providing background on the relevance of level of education and diabetes in relation to prostate cancer as with other contributing factors such as tobacco smoking.

I don't detect any flaws in the data analysis, interpretations or conclusions that would prohibit publication or require significant revisions. I believe the methodologies are sound and meets the standards of the field and would allow other researchers an opportunity to replicate their findings.

Overall, I feel that this manuscript is worthy of publication.

Reviewer's comments and Authors' response:

We greatly appreciate the constructive and very helpful comments by the four reviewers. Below is our response to the comments.

Reviewer #1 (Remarks to the Author): with expertise in prostate cancer, cancer immunology

For these studies, Minas et al performed obtained sera from men with prostate cancer who were African (Ghana cohort), African-American or European American. Serum proteomics were evaluated using the Olink technology, and a robust number of cases and controls were analyzed. The demographic of the pts from whom the samples came are described in the Materials and Methods section. The data are descriptive and a bit diffuse, but very interesting. Briefly they found that common demographic factors (age, BMI etc) associated differently with markers in different populations, that samples from men from Ghana clustered differentially from AA or EA samples, and they also validated some of these associations by quantifying genetic inheritance using genomic markers. That's very rarely done, and is a clear high point of the data. A signature associated with suppression of tumor immunity was associated with ACM and CM across pts. Finally, a reductionist two gene signature including only 41BB (TNFRSF9 and PTN) was strongly associated with PC mortality but only in AA pts. Although the results are largely descriptive, the number of patients, robust statistical analyses and novel findings make this an important addition to the literature, and as such I have only minor comments.

Response: The authors thank the reviewer for the supporting remarks.

Reviewer's comments with our response

Point #1. Some key data, i.e. the association between different proteomic markers is buried in a supplemental Figure and table (S3 and table S3). I would strongly advocate for including these data in the main manuscript. Only a handful of studies have shown how cytokines associate, and those data are mostly from Luminex or mesoscale data. To my knowledge, this is probably the first such proteomic analysis performed, or at least the first analysis in PC pts, and as such is probably worthy of a little more attention.

Response: We followed the advice and moved data from previous Supplementary Fig. 3 into the main manuscript, as the new main Figure 1. The new figure shows the correlation matrix for the 82 immune-oncology markers among African American controls and cases, as an example. We show only two matrixes in the main figure for better visibility of the data. We still maintain Supplementary Fig. 3, showing the correlation matrix for the immune-oncology markers among Ghanaian and European-American men, for both controls and cases.

Point #2. Results – line 97: The phrase “were detected” needs a bit more clarification in the results section. Does each analyte have its own lower limit of detection? What are they? If available, they could be listed in a supplemental table. Similarly, for figure 1 it wasn't clear to me whether data were being correlated as continuous variables, or in terms of present vs absent. This should be clarified in the results section.

Response: We agree with the reviewer that more clarification was needed. The phrase “were

detected” means the levels of the analytes were above their respective lower limit of detection. We have made this clear in the revised result section (on page 5) and added a new Supplementary Table 2 that lists the lower limit of detection for all 92 proteins assayed. In Figure 2, the analytes are being correlated as continuous variables. We have clarified this point in the revised figure legend and method section.

Point #3. Results – line 88: ‘Patient characteristics have been previously described’. The authors absolutely need to expound a bit on patient characteristics here, it is a bit presumptuous to assume that the reader is expected to go back and read all three prior papers just to understand this one. Should consider saying “Briefly, the NCI-Ghana study included xx men with yy disease with a median age of xx. The NCI-Maryland PCCC included xx ... etc. Just 3 additional sentences should do the trick.

Response: Per recommendation, we have added a brief paragraph at the beginning of the result section, on page 4, to describe the patient characteristics of both the NCI-Ghana and NCI-Maryland Prostate Cancer Case Control studies, and also to describe some key differences among the controls as many of the analysis were done with a focus on the control population.

Point #4. Results line 137- these extensive comments on CAIX are out of place here, they belong in the discussion section.

Response: We have removed the sentences that discuss CAIX from the result section.

Point #5. Figure 2 – unsupervised clustering. This figure doesn’t add much, other than to demonstrate that the Af pts cluster differentially as compared to the AA and EA pts. The row text is too small to read, and the AA and EA are quite interspersed. My suggestion would be to move to move this figure to supplemental data if that were required to include supplemental table 3a and figure 3 as above.

Response: We thank the reviewer for the comment. Yet, we think Figure 2 is important in that it visualizes one of our major findings that ancestral background may have a significant impact on the global immune-oncological protein profile. To address the reviewer concern that there may not be sufficient separation between AA and EA population controls, we have performed an additional, more formal analysis by dissecting the hierarchical clustering dendrogram to extract K clusters (with K=2, 3) across the three population groups and investigated the statistical significance of the resulting contingency tables using Fisher’s or chi-squared tests, as appropriate. We found significant differences in cluster representation between Ghanaian, African-American, and European-American men with cluster enrichment by population group at $P < 1.e-10$. The findings from this analysis have been added to the revised manuscript as new Supplementary Fig. 5 and are described in the result section, on page 8 (starting line 189):

We performed an additional statistical analysis of cluster assignments to more formally establish that the immune-oncological protein profile defined by the 82 markers is indeed different between these groups of men. We obtained the cluster assignments by cutting the hierarchical clustering dendrogram to extract K clusters (with K=2, 3) and tested for differences in their distribution across the population groups (Supplementary Fig. 5). We found significant differences in cluster representation between Ghanaian, AA, and EA men with cluster

enrichment by population group at $P < 1.e-10$, confirming that significant differences likely exist in the global immune-oncological protein profile among them.

Point #6. Figure 6 – These are key data. Would suggest including mention that TNFRSF9 is CD137 / 4-1BB in the legend for clarity. PTN also needs to be defined in the figure legend as pleiotrophin, my guess is at least some readers are going to be a bit confused as to whether there's some serum marker of PTEN signaling or loss?

Response: We have now added the two other names/abbreviations for TNFRSF9 and defined PTN as pleiotrophin in figure legends of Figure 7 and Supplementary Fig. 7 to avoid any confusion for the reader.

Point #7. Discussion – The discussion on TNFRSF9 section is well thought out. Here CD137 / 4-1BB should also be mentioned for reference. The discussion of pleiotrophin is far too short; can the authors please provide some additional information on the function and expression pattern of this analyte? They should also present at least one or two hypotheses on why PTN is associated with risk, and why it might co-associate with 41BB.

Response: We thank the reviewer for the comments. We added the two other names for TNFRSF9 in the discussion part, as suggested. We have also expanded the discussion related to pleiotrophin, as the reviewer recommended.

On pages 16 (starting line 377)-17, we added the following paragraph:

PTN is a secreted cytokine that is developmentally regulated. Normally expressed during embryogenesis as a growth or differentiation factor, it is expressed either at very low levels or not at all in healthy adults^{55,56}. PTN re-expression in adults is associated with tumor development, metastasis and angiogenesis with elevated expression reported in several cancer sites⁵⁷⁻⁵⁹. This increased expression of PTN has been associated with poor prognosis in colorectal cancer⁵⁷, hepatocellular carcinoma (HCC)⁵⁹ and gliomas⁵⁸. Several hypotheses have been pursued to find out how PTN exerts pro-metastatic effects. For example, it is proposed that PTN promotes cancer progression through increased vascular endothelial growth factor deposition at the vasculature leading to vascular disruption⁵⁸. PTN may upregulate lipid synthesis, contributing to hepatic stenosis and progression of HCC⁵⁹. Serum PTN is a candidate biomarker for occurrence of breast⁶⁰ and small cell lung cancers⁶¹ and has been shown to associate with metastatic prostate cancer⁶², consistent with the findings in this study. In prostate cancer, PTN regulates mesenchymal and epithelial proliferation, with PTN itself being regulated by the androgen receptor during prostate development⁶³. To the best of our knowledge, this is the first time a potential relationship between PTN and TNFRSF9 has been described. However, PTN has been implicated in the induction of TNF- α expression in peripheral blood mononuclear cells demonstrating a link between PTN and the TNF superfamily in the circulation⁶⁴.

Reviewer #2 (Remarks to the Author): with expertise in biostatistics, cancer

Minas and colleagues have undertaken a remarkable dataset generation, creating serum proteomes on several thousand men, approximately equally divided between prostate cancer

patients and controls. The dataset QC is extremely well-performed, and the resource will have very substantial value in and of itself.

Response: The authors thank the reviewer for the supporting remarks. As the reviewer will see with our responses to all raised points, we followed the advice and adjusted for multiple comparing testing in all analysis and report now FDRs. We also added an adjustment for disease presentation at diagnosis to the survival analysis, as requested. Notable, the previously reported association of the suppression of tumor immunity signature with prostate cancer-specific survival rather strengthened with this additional adjustment. Overall, these revisions clearly improved the quality of our manuscript.

1. Data Access

Point #1.

This dataset provides a major community resource. The authors have not indicated how all molecular and associated clinical and demographic data are available. That is central component of the value of this study.

Response: Thank you. To ensure full transparency, reproducibility, and dataset reusability, we created a GitHub repository to permanently store all molecular and associated clinical and demographic data used in this study, as well as our scripts to generate results and figures (see “Data and code availability” section). During the manuscript review process, we protect the confidentiality of our data and results by keeping the repository private. For reviewing purposes, we made available a cloned full copy of the GitHub repository that reviewers can access at the following Google Drive location:

https://drive.google.com/file/d/1D6vewhRI7enMI8igoTQeJdR_ea2D9hET/view?usp=sharing

2. Statistical Modeling

Point #2. 2a) The authors repeatedly incorrectly compare two features based on p-values within a single or different model. For example in lines 123-129 the authors incorrectly comparing models based on their p-values relative to the null. The correct approach is to compare the models directly, and state whether the coefficient for a gene is significantly different between the two cohorts: each model gives an effect-size (the coefficient) and its confidence interval. These need to be compared. Another example is lines 255-257 which compares AA to EA men. All statements of "exclusivity" of association to specific subgroups throughout need to be corrected. The two classic statistical approaches are directly comparing coefficients and their confidence-intervals (or other variance measure) as noted above, or by fitting a single multivariate model with stratification or other control and extracting the comparison of interest directly using a contrast matrix.

Response: We addressed this point and improved our analysis by implementing the approach of confidence interval comparison, explained in the Methods subsection “Association of clinical/socio-demographic characteristics with immune-oncological proteins”. We added a Supplementary Data 1 file with detailed regression results for each cohort and the analysis of statistical significance across cohorts. We have updated the paragraphs in the result section that describe Figure 2, pages 5-7, to reflect the updated analyses.

Point #3. 2b) Multiple testing is repeatedly ignored. For example, Figure 1 needs to be adjusted

for multiple-hypothesis testing, as do many other areas (e.g. Table S5, Figure 4, etc.). The authors sometimes note a correction, but also present unadjusted results. The lack of correction is sometimes fatal to conclusions being drawn. For example the authors highlight in Figure 5B that elevated suppression of tumor immunity is associated with PCSM. Unfortunately this is marginally significant ($p = 0.057$) and the result is not adjusted for multiple testing. In fact the majority of survival analyses appear not to survive multiple testing correction. Multiple-testing correction is not optional: it must be applied throughout, and systematically in a uniform way. And only adjusted results should be presented.

Response: Thank you. We now addressed this point throughout the manuscript, rigorously adjusting for multiple comparing testing. As the reviewer will see, most of our observations remain significant after the adjustment. We now provide FDR-adjusted data for each analysis that we performed. However, we don't think that the survival analysis would warrant multiple testing adjustment. We want to point out the ongoing discussion when and where adjustments for multiple comparison analysis is recommended. Highly cited commentaries by Kenneth Rothman (PMID: 2081237) and Andre Althouse (PMID: 27106412) discussed this issue, rejecting the concept of undifferentiated multiple testing adjustments. Most epidemiological reports, even in the top journals like NEJM and JAMA, don't have it, for a reason. Large scale omics studies and the GWAS approach certainly require it - but the analysis of survival that we performed would not fall in this category. The approach is rather *a priori* testing whether signature scores are associated with survival outcomes – per hypothesis one would test for all survival outcomes. Also, as the reviewer will see that the association of suppression of tumor immunity with lethal prostate cancer strengthened when we added the adjustment by disease presentation at diagnosis (using the NCCN risk score), as was recommended by this reviewer, and provide FDR-adjusted significance testing in Supplementary Tables 11, 13, and 14.

Point #4. 2c) The methods are unclear: was a separate model fit for each clinical/socio-demographic variable (with each protein), or was a single model fit for each protein with all variables included? If separate models were fit, then the authors need to update to a single model. If a single model was fit, then the methods need rewriting for clarity (presumably with an equation).

Response: A single model was fit for each protein with all variables included. This point was made clearer by adding further details (and the regression equation used) to the Methods subsection "Association of clinical/socio-demographic characteristics with immune-oncological proteins".

Point #5. 2d) Line 180-182, looking at dendrogram groupings is not an appropriate way of determining proximity of groups. A formalized statistical analysis is required using distance metrics. Further, a separate hypothesis is that there is a cohort-wise difference in sample collection, storage, extraction, or other handling. Have the authors excluded this possibility? If so, please outline more clearly in the text. If not this is a significant caveat to several of the analyses that needs to be clearly noted.

Response: An additional analysis was performed to address this point. The dendrogram (derived from performing hierarchical clustering of subjects based on the distance metric $d=1-r$, where r is Pearson's correlation) was cut to extract K clusters (with $K=2, 3$); then, Fisher's and

chi-squared tests were performed on the resulting contingency tables shown in the new Supplementary Figure 5. We found significant differences in cluster representation between Ghanaian, African-American, and European-American men with cluster enrichment by population group at $P < 1.e-10$.

On page 8 of the revised manuscript we write:

We performed an additional statistical analysis of cluster assignments to more formally establish that the immune-oncological protein profile defined by the 82 markers is indeed different between these groups of men. We obtained the cluster assignments by cutting the hierarchical clustering dendrogram to extract K clusters (with K=2, 3) and tested for differences in their distribution across the population groups (Supplementary Fig. 5). We found significant differences in cluster representation between Ghanaian, AA, and EA men with cluster enrichment by population group at $P < 1.e-10$, confirming that significant differences likely exist in the global immune-oncological protein profile among them.

We cannot exclude that there are cohort-wise differences related to the NCI-Ghana study although we are not aware of them based on the analysis or the protocols that were used. There would not be a cohort effect in the NCI-Maryland study because AA and EA men have been recruited into this study during the same time period, using the same recruitment tools/recruiters, the same hospitals, and the same collection, storage, extraction, or other handling protocols.

Yet, to address this point we added the following paragraph to the discussion, on page 18:

“Our study has strength and limitations. The major strength is the large sample size, the measurement of 82 immune-oncology markers with a robust technology, and the inclusion of men from Ghana and the U.S. Moreover, we applied multiple testing adjustments in reporting the significance of our observations. However, we collected blood samples in Ghana and the U.S. Although blood sample collection in Ghana followed a protocol that was following standards of practice in the U.S., we cannot exclude that serum preparation and shipping had an effect on the performance of the immune-oncology marker measurements, yet we are not aware of such an influence. In our survival analysis, we included 57 events for prostate cancer-specific deaths, 103 events for any cancer deaths, and 202 events for all-cause mortality among men with prostate cancer. In these analyses, we found that an elevated suppression of tumor immunity score to be consistently associated with decreased survival. Yet, additional studies will be needed to further confirm the association of the suppression of tumor immunity signature with lethal prostate cancer.”

Point #6. 2e) The fraction of variance explained by ancestry (Table S6) seems largely uninterpretable based on p-values. Because of the large sample-size, even tiny effect-sizes are statistically significant. For example, the authors are indicating that 0.6% of variance explained by ancestry is a statistically significant result. Statistically, yes, but the biological interpretability of that tiny fraction is not clear. Of the 45 statistically-significant hits, only 7 explain even 10% in

Table S6. Indicating p-values as a bright-line threshold independent of effect-size context is very poor statistical practice. Further, confidence intervals on these variance-explained estimates are required to interpret them appropriately.

Response: We appreciate the reviewer's comment. There is no accepted cutoff to determine whether a small variance explained will have a biological implication. This is the reason why we made all the data and statistics available to readers for them to decide. We now added the following sentence to the manuscript on page 9 (starting line 201):

"The approach showed that, to some extent, the variance in the levels of several immune-oncological analytes can be strongly influenced by the degree of West African ancestry of these individuals (Fig. 4A). The variance in 39 of the analytes were significantly (FDR-adjusted $P < 0.05$) influenced by degree of West African ancestry (Supplementary Table 6, Supplementary Data 2). The levels of 37 analytes were significantly accounted for by West African ancestry even after adjusting for age, BMI, aspirin use, education, income, diabetes, and smoking status (Supplementary Table 7, Supplementary Data 3). CXCL5, CXCL1, MCP2, MCP1, CXCL11, CCL23, PTN, TWEAK, NCR1, IL18 and CCL17 were the top-ranked proteins. West African ancestry contributed to the variance with various effect sizes and explained >10% of the variance among the top 7 proteins (Supplementary Tables 6-7, Supplementary Data 2-3)."

Point #7. 2f) In a similar way, the authors report Spearman's rho for chemotaxis and tumor immunity suppression (line 218). All analyses need to be from appropriate multivariate models, reporting effect-sizes (not simply correlations) and their confidence-intervals.

Response: Per recommendation of the reviewer, we now performed multivariable regression analysis, instead of a Spearman's correlation analysis, to assess the correlation of chemotaxis and suppression of tumor immunity with ancestry. On page 10 of the revised manuscript, we write as follows, starting line 233:

"Ghanaian men had even higher scores for chemotaxis and suppression of tumor immunity than both AA and EA men (Fig. 5C and E), indicating a possible association with West African ancestry. The latter was corroborated with our finding that the chemotaxis and suppression of tumor immunity scores positively correlated with the proportion of West African ancestry within the NCI-Maryland cohort, even after holding the other variables constant (i.e. age, BMI, education, aspirin use, diabetes, and smoking history) in the regression analysis (for chemotaxis score: regression coefficient= 5.12 (3.75, 6.49), $P < 0.0001$; for suppression of immunity score: regression coefficient=4.02 (2.01, 6.04), $P < 0.0001$). Even though apoptosis and vasculature-associated cytokines were not significantly different between EA and AA men, we found both processes to be elevated in the Ghanaian men."

Point #8. 2g) Survival models are intriguing, however the largest confounding factor is the state of the disease at diagnosis. The models do not appear to control for pre-treatment serum PSA, T-category, M-category, N-category or grade. The authors corroborate this concern by showing in the NCI-Maryland cohort that this score is preferentially associated with metastatic disease at diagnosis. So the conclusion appears to be "metastatic prostate cancer appears to occur in

patients with enrichment of this functional group". The survival analyses are simply a distraction from uncontrolled models that confuse this analysis.

Response: We thank the reviewer for the comment. We made this adjustment using the National Comprehensive Cancer Network (NCCN) Risk Score assignment that describes prostate cancer aggressiveness based on PSA, TNM stage, and Gleason grade at disease diagnosis. The NCCN classification is widely accepted and recommended by clinicians. Thus, cases were assigned to NCCN risk groups based on the patients' TNM stage, Gleason score, Gleason pattern, and PSA level at diagnosis according to the 2019 NCCN guideline for prostate cancer. The association of suppression of tumor immunity with survival adjusted for NCCN scores is now reported in the revised Figure 6 and Supplementary Tables 11, 13, and 14 with *P* values and *q* values. It turns out that the adjustment rather strengthens the association of the suppression of tumor immunity score with survival when compared to the model that did not contain this variable.

Point #9. 2h) Line 259 suggests that the authors aim to identify causal drivers, however none of the subsequent analyses indicate causality. The model presented in Figure 6 needs control for established clinical features associated with mortality (PSA, T, N, M, grade).

Response: We agree with the reviewer and have corrected the phrase in line 259 (initial submission, now on page 12, line 283) from "causal drivers" to "potential drivers". To address the reviewer's next comment, we now performed the elastic net Cox regression and the subsequent "traditional" Cox regression analysis after adding the NCCN score as a co-variable. In the elastic net Cox regression, 88 features and the NCCN risk score were evaluated as potential predictors. The NCCN risk score was the top predictor of the lethal disease, as one might expect. However, the previous top two analytes PTN and TNFRSF9 remained the most predictive features besides the NCCN risk score, and the three features combined predicted prostate cancer-specific mortality with 90% accuracy. We also added the NCCN risk score to the "traditional" Cox regression model, and as shown in Figure 7C, the combination of PTN and TNFRSF9 remained predictive of lethal prostate cancer after adjusting for it. Thus, the combination of the two markers remained independently associated with lethal prostate cancer.

3. Writing

Point #10. The discussion contains substantial speculation, and would benefit from trimming that and replacing with some detailed comments on the caveats of the study and analyses. The tone and wording of the study overall is too aggressive given the data at hand, and needs softening.

Response: Thank you. We revised the discussion, also in response to requests by other reviewers, and added a strength and limitation section to it, on page 18, as outlined in the response to point #7. In addition, we think that the inclusion of multiple testing adjustments in our analysis approach, as suggested by the reviewer, has further strengthened the validity of our observations.

Reviewer #3 (Remarks to the Author): with expertise in prostate cancer, cancer immunology

The authors have written a very nice paper in which they describe different findings in three different populations (Ghanaian, African American (AA) and European American men) and measured 82 circulating proteins in almost 3000 men with and without prostate cancer. In men of West African ancestry protein signature for immunosuppression was significantly elevated. Two markers, pleiotrophin and TNFRSF9 predicted poor survival in AA men. I am very enthusiastic about this paper and it contains novel information.

As a clinician with a special interest in immunotherapy, I consider the paper well written with an appropriate interpretation of the immunological data.

Response: The authors thank the reviewer for the supporting remarks.

I only have minor comments:

Point #1. The initial PSA in the Af population is 52, whereas the iPSA in the other two groups (AA EA) is 6.8 and 6.0. It is probably due to later detection of prostate cancer in Ghana. However, more advanced disease could be associated with a more immunosuppressive environment. Please discuss that your findings are linked to ancestry and not to more advanced disease.

Response: We thank the reviewer for the comment. We agree that differences in the immune-oncological proteins could be confounded by disease status, which is why we investigated the impact of ancestral background on the immune-oncological proteins among the healthy control populations. The results displayed in Figures 2-5 were obtained from healthy controls. Hence advanced disease/PSA level would not influence the observed trends. We show correlation data for cases only in the new Figure 1 and Supplementary Fig. 3, and here stratified for Ghanaian, AA, and EA men with prostate cancer.

Point #2. line 47, line 239, and line 535: You state in the paragraph about the statistical analysis that an association is only considered statistically significant if the P-value is less than 0.05. In your abstract and line 239 you are using phrasing like pointing to clinical significance and marginally significant. Please use the proper terms for significant and non-significant. I regret that in such a beautiful paper this misnomer is applied.

Response: We appreciate the reviewer's comment. We have removed such phrases from the manuscript.

Reviewer #4 (Remarks to the Author): with expertise in prostate cancer, health disparities

I don't detect any flaws in the data analysis, interpretations or conclusions that would prohibit publication or require significant revisions. I believe the methodologies are sound and meets the standards of the field and would allow other researchers an opportunity to replicate their findings. Overall, I feel that this manuscript is worthy of publication.

Response: The authors thank the reviewer for the supporting remarks.

Reviewer's comments with our response

Point #1. I think the work supports the conclusions. However, I believe the authors should devote more effort providing background on the relevance of level of education and diabetes in relation to prostate cancer as with other contributing factors such as tobacco smoking.

Response: Thank you very much for your comments. We added two recent references describing the association of these factors with prostate cancer. On pages 12-13 we write:

Of those patient features, education as a surrogate for socioeconomic status and health care access, BMI, smoking status, and aspirin use have been associated with the risk of lethal prostate cancer whereas the association of diabetes with prostate cancer outcomes is less certain^{32,33}.

Reviewers' Comments:

Reviewer #1:

Remarks to the Author:

The revised manuscript has been carefully edited and is now much clearer and more relevant, this will be a significant addition to the literature. Further, all of my concerns have been well addressed. I have no further comments or criticism.

Reviewer #2:

Remarks to the Author:

1. Data Access

While GitHub is a suitable repository for storing code, it is not an appropriate mechanism for storing clinical or molecular data. In fact there would be nothing preventing the authors from removing the data or restricting specific users. Further, a google drive is not a suitably de-identified way of sharing data during the review process. The data must be deposited in an appropriate public repository, such as dbGaP or GEO, and a reviewer accession provided. The data is a critical component of the value of this study.

2. Statistical Modeling

a) The authors now move to a simple comparison of the confidence interval containing zero (using an appropriate pooled SE estimate). This is not an appropriate procedure because it fails to correct for multiple-testing: it's effectively applying an unadjusted bright-line threshold. A formal statistical comparison of the effect-sizes is both standard in the field and needed here.

b) The authors have unfortunately elected not to control for multiple testing throughout, and this is incorrect. The authors define six biological pathways, systematically test each of these, and treat the single pathway that is significant independent of the multiple testing of the other five pathways. A procedure with a 5% false positive rate done six times does not have a 5% false-positive rate. Indeed the two cited papers suggest avoiding uniform "unconsidered" multiple-testing adjustment, and Althouse closes: "My colleagues and reviewers, in the future, I implore you to carefully consider whether multiple-comparison adjustment is truly necessary on a case-by-case basis." The claims being made in the unadjusted analyses are clear exploratory analyses of the complete set of pathways considered in the study. Adjustment is both typical in the field and required.

Reviewer's comments and Authors' response:

We thank the reviewer for the comments and addressed them with our revisions.

Reviewer #2 (Remarks to the Author): with expertise in biostatistics, cancer

Point #1. Data Access

While GitHub is a suitable repository for storing code, it is not an appropriate mechanism for storing clinical or molecular data. In fact there would be nothing preventing the authors from removing the data or restricting specific users. Further, a google drive is not a suitably de-identified way of sharing data during the review process. The data must be deposited in an appropriate public repository, such as dbGaP or GEO, and a reviewer accession provided. The data is a critical component of the value of this study.

Response: Thank you for your comments. We deposited the clinical and molecular data for this study at the Open Science Framework, one of the repositories approved and recommended by Nature. To protect the confidentiality of our data, the repository will remain private while the manuscript is under review. Reviewers may access the private repository at

<https://osf.io/327ha/>

using the following anonymized read-only link:

https://osf.io/327ha/?view_only=ce91ea5c75f94f38bc1f0b419c3467c4

In addition, the data will remain in the GitHub repository (alongside with the code) to ensure the transparency and reproducibility of our results. This repository will remain private while the manuscript is under review. Both repositories (OSF and GitHub) will be made permanently public without any access restrictions upon publication of our manuscript.

Point #2. Statistical Modeling

- a) The authors now move to a simple comparison of the confidence interval containing zero (using an appropriate pooled SE estimate). This is not an appropriate procedure because it fails to correct for multiple-testing: it's effectively applying an unadjusted bright-line threshold. A formal statistical comparison of the effect-sizes is both standard in the field and needed here.
- b) The authors have unfortunately elected not to control for multiple testing throughout, and this is incorrect. The authors define six biological pathways, systematically test each of these, and treat the single pathway that is significant independent of the multiple testing of the other five pathways. A procedure with a 5% false positive rate done six times does not have a 5% false-positive rate. Indeed the two cited papers suggest avoiding uniform "unconsidered" multiple-testing adjustment, and Althouse closes: "My colleagues and reviewers, in the future, I implore you to carefully consider whether multiple-comparison adjustment is truly necessary on a case-by-case basis." The claims being made in the unadjusted analyses are clear exploratory analyses of the complete

set of pathways considered in the study. Adjustment is both typical in the field and required.

Response:

Point #2a: In order to assess the statistical significance of cohort differences in the association between analytes and clinical/socio-demographic characteristics, we fitted each analyte to a model with interaction terms according to the formula “analyte ~ age*Gh + bmi*Gh + education*Gh + aspirin*Gh + smoking*Gh + diabetes*Gh + PSA*Gh + age*Afr + bmi*Afr + education*Afr + aspirin*Afr + smoking*Afr + diabetes*Afr + PSA*Afr”, where the intercept and non-interacting terms were implicitly also included in the model. Here, two dummy variables were introduced: Gh (defined as 1 for Ghanaian subjects, 0 otherwise) and Afr (defined as 1 for Ghanaian and African American subjects, 0 for European Americans). Models were selected based on significance of the FDR-adjusted P value <0.05 on the F-statistic; if this condition was satisfied, the association between the target analyte and each individual covariable was characterized by the corresponding FDR-adjusted P value and coefficient.

Full results are now provided in the 4th tab of Supplementary Data 1. The discussion following Figure 2 has been modified accordingly, highlighting the significant interactions found from this modeling approach. We updated the “Association of clinical/socio-demographic characteristics with immune-oncological proteins” subsection in the Methods section to reflect these additions.

Point #2b: We thank the reviewer for the comments. However, we would like to clarify our analysis approach. What is being shown in Figure 6 are the data from three multivariable survival analyses related to all-cause mortality, prostate cancer-specific survival, and death from any cancer following a prostate cancer diagnosis. In figure 6, we highlight the association of the six immune-oncology marker-defined pathways as covariables with patient survival. The analysis shows that only suppression of tumor immunity is associated with the three survival outcomes independent of the other covariables – none of the other pathways. We did not perform additional analyses – just one multivariable analysis per survival outcome. With the previous resubmission, we added the NCCN risk score as a covariable, as was recommended by this reviewer, to account for differences in disease presentation at time of diagnosis in these analyses. The now revised figure 6 shows hazard ratios related to the six pathways with the NCCN risk scores in the model but now with the 99% confidence interval for each pathway (instead of the 95% CI), to strengthen the findings. We also provide multi-testing adjustments of the P values in Supplementary Tables 11-14. We hope these changes address the reviewer’s concern. We also made clear in the text that this survival analysis is hypothesis generating, pointing to a possible distinct role of suppression of tumor immunity in prostate cancer survival that should be tested in other patient cohorts.

Reviewers' Comments:

Reviewer #2:

None